# Selective knockout of murine glutamic acid-rich protein 2 significantly alters dark continuous noise in rod photoreceptors

Delores A. Stacks[1], Ulisse Bocchero[2], Marci L. DeRamus[1], Mai N. Nguyen[1], Jeffrey Messinger[1], Timothy W. Kraft[1], Steven J. Pittler[1] and Johan Pahlberg[2,3]

[1] *Department of Optometry and Vision Science, Vision Science Research Center, University of Alabama at Birmingham, Birmingham, AL, USA*
[2] *Photoreceptor Physiology Group, National Eye Institute, National Institutes of Health, Bethesda, MD, USA*
[3] *Photoreceptor Physiology Group, National Eye Institute, National Institutes of Health, Porter Neuroscience Research Center, Bethesda, MD, USA*

Handling Editors: Katalin Toth & Karin Dedek

The peer review history is available in the Supporting information section of this article (https://doi.org/10.1113/JP286548#support-information-section).

**Abstract figure legend** GARP2 has a minor role in maintaining the structural integrity of the rod outer segments as a function of age, but an important role in regulating rod photoreceptor continuous noise, likely through stabilization of PDE6 basal activity.

**Abstract** Glutamic acid-rich protein 2 (GARP2), a glutamic-acid-rich protein found exclusively in rod photoreceptors, has been suggested to function as a structural protein, a modulator of the cGMP enzyme phosphodiesterase type 6 (PDE6), and a gating inhibitor of the rod cGMP-gated cation channel. GARP2 is a splice variant of the *Cngb1* gene, which in the rods encodes the $\beta$-subunit of the cyclic nucleotide-gated cation channel. Mutations in *Cngb1* cause retinitis pigmentosa (RP45),

D. A. Stacks and U. Bocchero share first authorship.

This article was first published as a preprint. Stacks DA, Bocchero U, DeRamus ML, Nguyen MN, Messinger J, Kraft TW, Pittler SJ, Pahlberg J. 2025. Selective knockout of murine glutamic acid-rich protein 2 (GARP2) significantly alters cellular dark noise in rod photoreceptors. bioRxiv. https://doi.org/10.1101/2025.05.09.653122

and $\beta$-subunit knockout mice are studied as models of this disease. In this work, using zinc finger nuclease-mediated gene editing, we have selectively eliminated GARP2 expression, while not affecting expression of the cyclic nucleotide gated cation channel $\beta$-subunit, to determine its essential functions in mouse rods. The absence of GARP2 caused no consistent perturbations of retinal structure. Transiently, rod outer segment length was regionally greater than wild-type and infrequently misaligned, appearing parallel to the retinal pigment epithelium. Electroretinography of the knockout mice did reveal consistent functional alterations over time, seen as a reduction in the electroretinography response amplitudes in older mice, albeit with no significant alterations in sensitivity to light. Interestingly, single-cell patch-clamp recordings showed a significant reduction in rod photoreceptor dark noise consistent with a previously proposed role for GARP2 in binding to PDE6 and affecting its basal activity. Our results suggest a role for the GARP2–PDE6 interaction in stabilizing the PDE6 enzyme and controlling the turnover rate of cGMP in darkness, adjusting the level of dark noise and implicating an influence on the signal and noise properties of rod photoreceptors.

(Received 8 April 2024; accepted after revision 29 October 2025; first published online 27 November 2025)

**Corresponding authors** S. J. Pittler: Department of Optometry and Vision Science, Vision Science Research Centre, University of Alabama at Birmingham, Birmingham, AL 35294, USA. Email: pittler@uab.edu

J. Pahlberg: Photoreceptor Physiology Group, National Eye Institute, National Institutes of Health, Bethesda, MD 20892, USA. Email: johan.pahlberg@nih.gov

**Key points**

- Glutamic acid-rich protein 2 (GARP2), an alternatively spliced variant of the *Cngb1* gene, is exclusively expressed in rod photoreceptors, but the *in vivo* role of GARP2 remains unestablished.
- We used precision gene editing technology to selectively knockout soluble GARP2 expression in rods, to determine its essential roles in structure and function of the retina.
- We show that GARP2 has a minor role in maintaining the structural integrity of the rod outer segments as a function of age.
- Our study indicates that GARP2 has an important role in regulating rod photoreceptor continuous noise, likely through stabilization of phosphodiesterase 6 (PDE6) basal activity to maintain the appropriate cGMP turnover rate.
- This regulation is critical to facilitate the single-photon sensitivity and function of the rod photoreceptors.

## Introduction

The rod photoreceptor is the most abundant cell type in the neural retina of nocturnal animals. During phototransduction, a photon of light is absorbed by a molecule of rhodopsin found on the membranous disks of the rod outer segment (ROS), activating the visual phototransduction signalling cascade and ultimately

**Delores Stacks** is currently working as a senior researcher and laboratory manager in the Department of Molecular and Cellular Pathology at the University of Alabama at Birmingham (Birmingham, AL, USA). This work was a part of her doctoral dissertation awarded in July 2018 under the mentorship of Dr. Steven J. Pittler from the University of Alabama at Birmingham, School of Optometry, Vision Science Graduate Program. **Ulisse Bocchero** received his Masters degree in Molecular and Cellular Biology from the University of Turin, Italy, and his Ph.D. in Neurobiology from the International School for Advanced Studies, Trieste, Italy. After completing his doctoral degree, he has been a Postdoctoral Fellow at the National Eye Institute (NIH, Bethesda, Maryland, USA) in the Laboratory of Dr. Johan Pahlberg since 2019, working on photoreceptor physiology, retinal circuitry function, and vision at the limits of detection.

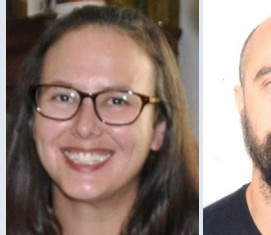
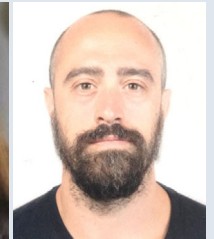

closing the cyclic nucleotide-gated (CNG) channel on the plasma membrane. The CNG channel is a heterotetramer composed of three $\alpha$- and one $\beta$-subunits (Weitz et al., 2002; Zheng et al., 2002; Zhong et al., 2002). The channel has sites for cGMP binding (Körschen et al., 1999), a calcium calmodulin-binding site (Grunwald et al., 1998), a transmembrane pore region, a gating region, and the glutamic acid-rich N-terminal tail of the $\beta$-subunit (Barret et al., 2022), which contributes to the structure of the photoreceptor by linking the external plasma membrane to the disk membrane via its interactions with *rds*/peripherin-2 (Gilliam et al., 2012; Pearring et al., 2021).

Glutamic acid-rich proteins (GARPs) are alternatively spliced variants of the *Cngb1* gene (Ardell et al., 2000). Although a shorter version of the CNG channel $\beta$-subunit (without the N-terminal glutamic acid-rich region) is expressed in many other tissues throughout the body (such as the adrenal gland, brain, testis and colon), the GARP proteins are exclusively produced in rod photoreceptors, with GARP2 having the highest protein expression level of the three proteins (Colville & Molday, 1996; Körschen et al., 1999). The GARP proteins are intrinsically disordered (Batra-Safferling et al., 2006), allowing them to change their structure and possibly function depending on the cellular microenvironment (Dunker et al., 2008). Maintenance of ROS structure has been attributed to GARP2 and/or the N-terminal GARP region of the cGMP-gated cation channel via its interactions with *rds*/peripherin-2 (Pearring et al., 2021; Poetsch et al., 2001), and *in vitro* analysis of GARP2 has shown that it stoichiometrically binds to phosphodiesterase 6 (PDE6) in the dark (Pentia et al., 2006). It was shown that this interaction stabilized the basal activity of PDE6 in darkness, and it was suggested this may facilitate the single-photon response of the rods by preventing spontaneous release of the inhibitory $\gamma$-subunits of PDE6, thereby regulating the fluctuations in cGMP concentration and levels of dark-noise. Studies have also indicated that GARP2 and the N-terminal GARP region of the channel $\beta$-subunit can act as gating inhibitors of the channel, which may reduce current noise associated with channel openings in the absence of cGMP (Michalakis et al., 2011). While the function of GARP2 has been characterized via *in vitro* assays, the *in vivo* role of GARP2 remains unestablished.

We previously characterized a murine transgenic GARP2 overexpression model (GARP2-Ox) and found that *in vivo* overexpression alters the scotopic photoresponse leading to an increase in phototransduction gain and a slowed recovery time after exposure to a light stimulus, along with a mild reduction in overall ROS length compared to wild-type (WT) (Sarfare et al., 2014). We also found that the overexpression leads to an accelerated rate of degeneration, similar to what was observed in an *rds*/peripherin-2 mouse model (Chakraborty et al., 2015; Connell et al., 1991). In this study, we examine a selective murine knockout of GARP2 and assess the structural and functional changes within the retina. We show that ablating GARP2 expression, leaving the $\beta$-subunit expression intact, has a minor role in maintaining the structural integrity of the retina and rod outer segments, to prevent degeneration. Importantly, we show for the first time that GARP2 participates in controlling proteins of rod phototransduction and cellular dark-noise levels through the regulation of PDE6 and cGMP turnover rate. This activity may be critical to facilitate the exquisite sensitivity and function of the rod photoreceptor.

## Methods

### Animal care

Homozygous GARP2-KO and WT mouse lines were housed and cared for by the Animal Resources Program at UAB following UAB IACUC guidelines and the ARVO statement on the use of laboratory animals, and at NIH at the NINDS Animal Health and Care Section following guidelines of the Animal Care and Use Committees of the NIH (ASP 1344-18). Each line was bred into a common C57BL/J background through at least five generations at NIH, at the NINDS Animal Health and Care Section before use in any experiments. All experiments were thus performed on the same mouse strains. Mice were housed in 12/12 h light–dark cyclic lighting, in standard rodent cages, on a standard chow diet.

### Zinc finger nuclease-mediated knockout of GARP2

We worked with Sigma Chemical Co. (St Louis, MO, USA) to utilize their CompoZR™ zinc finger nuclease (ZFN) technology to identify the best deletion site near the targeted GARP2-specific exon 12a at the *Cngb1* locus 5′-TGG ACA AGC ATT GTC nnn nnn ACT GGG GTT GTA GGA TGG A-3′ with the 'n' representing the requisite 4–7 bp space between zinc finger targets to allow proper folding for *Fok*I restriction enzyme cleavage. Nucleotide BLAST analysis from NCBI (Altschul et al., 1990, 1997; Boratyn et al., 2012; Camacho et al., 2009; Madden et al., 1996; Morgulis et al., 2008; States & Gish, 1994; Zhang & Madden, 1997; Zhang et al., 2000) was used to ensure ZFN specificity to exon 12a. Exon 12a encodes the last eight amino acids of GARP2 and this exon is unique to soluble GARP2; thus removal of this exon selectively eliminates expression of GARP2, but does not affect expression of GARP1 or the *Cngb1* $\beta$-subunit. The UAB Transgenic and Genetically Engineered Models Facility injected mRNA encoding the ZFN into the

male pronucleus of fertilized eggs. One of 14 potential founder mice was identified through PCR genotyping with different deletions on each allele. Sanger sequencing was performed using the primers: sense – 5′-GGG GTG GTG GTG AAT GTC CTT-3′; and antisense – 5′-CTT TAC AAA GAC TAC TCT GGG GTT GAG C-3′. It was established through subsequent crosses to WT C57BL/6 mice that each allele was separable and germ-line transmissible. The allele with the longer deletion was bred to homozygosity and used for this study.

### PCR confirmation of KO and genotyping

Genotyping was performed on DNA extracted from mouse tail tissue using a previously described extraction method (Wang, & Storm, 2006). PCR was performed using Epicentre FailSafe PCR premix and primers flanking the region outside of exon 12a, which was the ZFN target region, using sense 5′-GGG GTG GTG GTG AAT GTC CTT-3′ and antisense 5′-CTT TAC AAA GAC TAC TCT GGG GTT GAG C-3′ primers. A PTC-200 thermal cycler (MJ Research-GMI, Ramsey, MN, USA) was set to the following parameters: one cycle at 94°C for 5 min; 33 cycles of 94°C for 45 s, 61°C for 45 s, and 72°C for 105 s; one cycle of 72°C for 7 min; and a post-run hold temperature of 4°C until removal from the instrument. A PCR product of 2.15 kb was obtained from WT mice, and a 1.34 kb product was observed in the GARP2-KO sample.

### Western blot and protein quantification

To confirm the ablation of GARP2, protein lysates were prepared from whole retinas of age-matched WT and GARP2-KO mice using 10 mM Tris–HCl and 0.1% Triton X-100. Protein concentrations were determined using Bio-Rad Protein Assay (Bio-Rad Laboratories, Hercules, CA, USA, cat. no. 500-0006). Thirty micrograms of each lysate were then separated on a 10% Mini-protean TGX gel and transferred to a polyvinylidene difluoride (PVDF) membrane using the Trans-Blot Turbo transfer system (Bio-Rad). Membranes were incubated with affinity-purified rabbit polyclonal antibodies against *Cngb1* N-terminus (Zhang et al., 2009), and mouse $\beta$-actin (1:1000 Sigma-Aldrich cat. no. A2228) primary antibodies overnight at 4°C, then incubated with corresponding fluorescent secondary antibodies (goat anti-rabbit 680, goat anti-mouse 800) for 1 h at room temperature.

For analysis of PDE6 subunits, retinas were excised from ~4.5-month-old WT and GARP2-KO mice and were individually lysed in 130 μl of RIPA buffer (Cell Signaling Technology, Danvers, MA, USA, cat. no. 9806S) containing 1.3 μl of Halt Protease/Phosphatase Inhibitor mix (Thermo Fisher Scientific, Waltham, MA, USA, cat.

no. 78442) in a TissueLyzer LT instrument, (Qiagen, Germantown, MD, USA). Protein was quantified in each homogenate using a Protein Assay (Bio-Rad Laboratories, cat. no. 500-0006). Separation of 20 μg of retina protein was done in a Criterion, 4–12% polyacrylamide Bis–Tris Gel (Bio-Rad, cat. no. 3450123) and transferred overnight to an Immobilon PVDF membrane (Millipore, Billerica, MA, USA, cat. no. IPFL00010,) using a wet transfer apparatus (Mini Protean 3 Cell, Bio-Rad). Blots were incubated with primary antibody against rabbit PDE6$\alpha$ (1:1000; ABclonal, Woburn, MA, USA, cat. no. A7915), rabbit PDE6$\gamma$ (R4842, 1:1000; a gift from Dr Ted Wensel, Baylor College of Medicine), and mouse $\beta$-actin (1:1000; Sigma-Aldrich, cat. no. A2228), followed by co-incubation with the corresponding secondary antibody (Thermo Fisher Scientific Alexa Fluor 700 goat anti-mouse, cat. no. A21036 and Alexa Fluor 800 goat anti-rabbit, cat. no. A32735). All blots were imaged, and protein quantification was done using the LI-COR Odyssey Quantitative Fluorescence Imaging System (Li-Cor Biosciences, Lincoln, NE, USA). PDE6 subunit statistical analysis was performed using a two-tailed Student's *t* test after removing one outlier using a *Z*-test.

### Light and transmission electron microscopy

Mouse eyes were enucleated following euthanasia with 5% isoflurane followed by cervical dislocation. The eyes were oriented by superior temporal corneal cauterization and were processed by the osmium and paraphenylenediamine (OTAP) method (Curcio et al., 2011). Tissues were postfixed in 1% OsO4 0.1 M in sodium cacodylate buffer for 2.5 h (dark, room temperature) with 6 min of cyclical push-processing in a microwave oven (Ted Pella, Redding, CA, USA), followed by 30 min in 1% tannic acid (Gallo tannin), a 50% acetonitrile wash (ACTN; EMS, Hatfield, PA, USA), and fresh paraphenylenediamine in 70% ACTN for 30 min. Dehydration steps with increasing ACTN concentrations were also push-processed. To improve resin infiltration (EMBed 812; EMS), we used the catalyst benzyl dimethylamine (EMS) and kept tissue on a rotating wheel for 3 days, replacing the resin three times. Tissues were cured in labelled flat embedding moulds (Ted Pella) for 24 h at 70°C. Sub-micron sections cut at 0.9 μm thickness (Diatome Histo knife, 6-mm edge; EMS) were obtained using a Leica Ultracut Ultramicrotome (Leica Microsystems Inc., Deerfield, IL, USA) into a reservoir of deionized water, warmed by proximity to a soldering iron to remove compression, placed on Fisherbrand™ Tissue Path Superfrost™ Plus Gold Slides (Thermo Fisher Scientific, cat. no. 22-035813) subbed in 0.01% poly-L-lysine (Sigma-Aldrich, cat. no. P8920) and dried at 40°C. The slides were stained with 2% toluidine blue O for

3 min, rinsed, dried, cleared in xylene and cover slipped (Permount; Fisher Scientific, Pittsburgh, PA, USA). Images were collected with an Olympus VS-120 microscope (BX61VS platform) running VS-ASW-2.9 software. Ultrathin sections (80–90 nm) were mounted on copper grids and visualized with a JEOL 1200 electron microscope equipped with a four-megapixel digital camera (AMT Gatan, Inc., Pleasanton, CA, USA, ES1000-785). Measurements of outer segment length and width and disk rim spacing were performed in ImageJ (NIH, Bethesda, MD, USA) (Abràmoff et al., 2004; Schneider et al., 2012). All reagents used for processing and staining tissue for imaging were acquired from electron microscopy services.

## Electroretinography

A custom-made electroretinogram rig (Visual Function Core, Dr Haouhua Quian, National Eye Institute, NIH) was used to assess light responses from live anaesthetized mice at postnatal month (PM) 3, 6 and 9, as previously described (Campla et al., 2022). All procedures were performed in dim red light. Mice were dark-adapted overnight and anaesthetized by intraperitoneal injection of ketamine (80 mg/kg) and xylazine (8 mg/kg). Pupils were dilated with a drop of a solution containing phenylephrine ophthalmic (1%), proparacaine HCl (0.5%) and tropicamide (1%) on the cornea. A second drop of 2.5% hypromellose ophthalmic demulcent solution was used to maintain moisture of the eyes. Mice were transferred to a heating pad (37°C) and electroretinography (ERG) responses were recorded from both eyes using gold wire loop electrodes placed over each cornea. Dark-adapted ERG responses were obtained using 20 ms light flashes of increasing light intensities (0.001–120 cd s/m$^2$). Responses were filtered at 50 Hz and sampled at 2 kHz. Each light response was averaged from 3–4 recordings every 10–90 s, depending on stimulus intensity.

## Single cell electrophysiology

A custom-made single-cell patch clamp rig (Laboratory of Dr Johan Pahlberg, National Institutes of Health) was used to assess the physiological properties of single rod and rod bipolar cells (RBCs) at PM3, 6 and 9, as previously described (Beier et al., 2022; Pahlberg et al., 2017). Whole-cell patch-clamp recordings were performed in retinal slices from mice dark-adapted overnight. All subsequent procedures were performed in darkness or under infrared (IR) illumination ($\lambda_{max} \sim 940$ nm). Mice were euthanized according to guidelines approved by NIH (ASP 1344-18), eyes were enucleated, and the retina was embedded in low-density agar (3%) in HEPES-buffered Ames' medium (pH 7.4). Retinal slices were cut at 200 μm

on a vibrating-blade microtome (Leica VT1000S) and visualized on the electrophysiology rig using an IR light source. The slices were super fused with Ames' medium (flow rate 8 ml/min), equilibrated with 5% $CO_2$–95% $O_2$, and temperature was maintained at 34–36°C (Warner Instruments, Hamden, CT, USA, TC-344C, SH-27B). The internal solution for whole-cell patch-clamp recordings contained (in mM): 125 potassium aspartate, 10 KCl, 10 Hepes, 5 *N*-methyl glucamine-HEDTA, 0.5 $CaCl_2$, 1 ATP-Mg, and 0.2 GTP-Mg; pH was adjusted to 7.4 with *N*-methyl glucamine-OH. Light-evoked responses were recorded from single rods and RBCs in the voltage-clamp mode; pipette resistance was ~15–17 MΩ, and holding potential was −40 and −60 mV, respectively. Light responses were obtained by delivering 20 ms flashes from a blue-green LED ($\lambda_{max} \approx 505$ nm, Thorlabs, Newton, NJ). Flash strengths varied from those producing a just-measurable response to those producing a maximal response, increasing from flash to flash by a factor of 2. Membrane voltages were recorded in current-clamp mode and corrected for junction potential (−10 mV). All responses were low-pass filtered at 300 Hz and sampled at 10 kHz.

## Electrophysiological data analysis

The *in vivo* ERG and single-cell data were analysed with Microsoft Excel, IGOR pro 8.04 (WaveMetrics, Lake Oswego, OR, USA) and a custom MATLAB (MathWorks, Natick, MA, USA) software package to extract parameters for maximum response amplitude ($I_{max}$), time to peak ($T_{peak}$), resting membrane voltage ($V_m$), and curve fitting for sensitivity and phototransduction gain. The analysis code is available at https://github.com/PahlbergLab.

Light sensitivity for each mouse genotype and each cell type was estimated from the half-maximal flash strength, which thus is independent of the maximum light amplitude current. Sensitivity ($I_{0.5}$) was calculated from the best-fit Hill equation to the intensity-response curve:

$$R/(R_{max}) = \left\{ 1/1 + \left( \frac{I_{0.5}}{I} \right)^n \right\}$$

where $R$ is the response amplitude, $R_{max}$ is the maximum response amplitude, $I$ is flash intensity in $R^*$/rod, $I_{0.5}$ is half-maximal flash intensity and $n$ is the Hill coefficient.

Phototransduction gain was estimated from average flash families. The Lamb–Pugh model of phototransduction was used to fit the leading edge of rod flash families to determine gain (Baylor et al., 1984; Breton et al., 1994; Hood & Birch, 1993; Lamb & Pugh, 1992; Pugh & Lamb, 1993):

$$(t) = \left\{ 1 \exp\left[ -0.5 \times \Phi \times A_g \times \left( t - t_{delay} \right)^2 \right] \right\} \times R_{max}$$

where $\Phi$ is the number of photoisomerizations per rod per flash (see above for calculation), $t_{delay}$ is the brief delay for the events of the phototransduction cascade to begin, and $A_g$ is the mathematically derived rate of phototransduction gain that focuses on PDE6 activation and CNG channel closure. Dark noise from rod photoreceptors was recorded in whole-cell voltage-clamp mode with a holding potential of $-40$ mV. *Continuous* noise was assessed from patch-clamp recordings from rod photoreceptors in complete darkness without any light stimuli, followed by a saturated bright light stimulus to close all transduction channels. To obtain the true cellular dark noise we subtracted the light adapted instrumental noise from the total dark noise recorded. We then constructed power spectra from the cellular dark noise by Fast Fourier transformation (FTT) on the traces and calculating the total power of the noise between 0.6 and 10 Hz.

All electrophysiology figures were made using Igor 8.0. Analysis code is available at https://github.com/ PahlbergLab.

## Statistical analysis

The statistical analysis for ERG and single cell parameters was performed with two-way ANOVA with Bonferroni correction in R (R Foundation for Statistical Computing, Vienna, Austria). The statistical analysis for the power spectral density (PSD) was done using a non-parametric test (Kruskal–Wallis) in R. The statistical analysis for PDE6 subunit expression and morphometric measurements was performed with unpaired, two-tailed, $t$ test in Microsoft Excel. Statistical significance is shown in the main text and in figure legends, defined as $* =$ p<0.05, $** =$ p<0.01, $*** =$ p<0.001. Data are reported as means $\pm$ standard deviation (SD), number of mice, retinas or cells depending on the experiments ($n$).

## Results

### GARP2-KO confirmed by PCR genotyping and western blot analysis

To knockout GARP2 expression, Exon 12a of the *Cngb1* gene was precisely targeted for deletion, as it encodes the last eight C-terminal amino acids unique to GARP2. Zinc finger nuclease (ZFN) technology was used to create this knockout, as it creates targeted double-strand breaks in genomic DNA using the *Fok*I endonuclease. Targeting is based on the alignment and subsequent folding of the two zinc fingers designed to recognize and bind to a specific 15–20 bp target per finger when those targets are separated by 4–7 bp (Fig. 1*A*). Two different deletion alleles were identified within the same founder mouse (Fig. 1*B*): one a complete deletion of Exon 12a (811 bp deletion, $\Delta$811),

and the other a deletion of the splice acceptor site and most of the 3′-untranslated region (503 bp deletion, $\Delta$503). The knockout was confirmed by Sanger DNA sequencing and PCR genotyping using oligonucleotide primers specific to regions outside the ZFN target site. Western blot analysis confirmed that the protein expression of GARP2 was ablated in our ZFN-mediated knockout, while WT has a distinct band at 73 kDa when immunoblotted with an anti-GARP2 antibody (Fig. 1*C*). The high number of negatively charged glutamic acid residues causes the predicted 31.9 kDa GARP2 protein to appear larger on a western blot, as previously reported (Batra-Safferling et al., 2006). To assess possible variation in the expression of the CNG $\beta$-subunit, we performed western blot analysis using 7-month-old WT and GARP2-KO mice. Retinal homogenates were separated by molecular mass, and we used an N-terminal specific $\beta$-subunit antibody that recognizes an epitope common to GARP2, GARP1 and the $\beta$-subunit. As shown in Fig. 1*D*, the $\beta$-subunit is readily detected in both WT and mutant (KO) retinas. GARP2 is observed in both WT samples but not in the mutant, and GARP1 is seen at low intensity in the mutant but not clearly discerned in WT, consistent with the poor detection of GARP1 by this antibody. Thus, our GARP2-KO is a true null allele and has selectively removed expression of GARP2 while retaining expression of GARP1 and the $\beta$-subunit. Since GARP2 has been shown to interact with PDE6 and prevent spontaneous activation of the inhibitory $\gamma$-subunits (Pentia et al., 2006), western blot analysis was performed to assess PDE6 expression in our KO mice (Fig. 1*E* and *F*). The expression level of PDE6A was comparable to WT (WT $n = 9$ retinas; GARP2-KO $n = 9$ retinas), whereas PDE6G expression was downregulated (WT $n = 8$ retinas; GARP2-KO $n = 8$ retinas), suggesting a role for GARP2 in stabilizing the PDE6 complex, likely through interaction with PDE6G, and in ensuring proper expression levels of the enzyme subunits.

### Structural changes in GARP2-KO

To assess any structural changes of the GARP2 deletion, we compared the morphology of GARP2-KO and WT retinas (Table 1). No major degenerative changes in overall structural integrity of the GARP2-KO retinas were observed at PM3, compared to similarly aged WT retinas (Fig. 2*A* and *B*, $n = 3$ retinas for both genotypes). At PM9 we observed a non-significant reduction of the INL (WT, $49.5 \pm 1.7$ μm, $n = 5$ retinas; GARP2-KO, $42.5 \pm 1.8$ μm, $n = 5$ retinas; $P = 0.13$), and the IPL (WT, $62.1 \pm 2.6$ μm, $n = 5$ retinas; GARP2-KO, $52.7 \pm 1.8$ μm, $n = 5$ retinas; $P = 0.06$), indicating no consistent perturbations of retinal structure. However, microscopically, at PM9 regions were found infrequently across the GARP2-KO

retina where the rod outer segments were longer (Fig. 2*C* and *D*). Comparing the means of the measurements, the GARP2-KO has an increased ROS length of about 1 μm, or 3.75% compared to WT (WT, 25.08 ± 2.67 μm, $n = 207$ rods, three retinas; GARP2-KO, 26.02 ± 5.09 μm, $n = 133$, three retinas; $P = 0.03$). A histogram of ROS length in 2 μm increments plotted against the normalized frequency of occurrence showed a normal monophasic distribution and a Gaussian fit of the WT histogram with a peak at 27.5 ± 0.08 μm (Fig. 2*E*). The ROS histogram for the GARP2-KO revealed a noticeable biphasic distribution (Fig. 2*F*), and a multipeak Gaussian fit of the GARP2-KO histogram found two distribution peaks at 26.8 ± 0.78 μm and 34.0 ± 2.6 μm. When the total area of the fit was divided by the area of the two peak families, 23% of the GARP2-KO ROS were longer than WT.

Thus, selective knockout of GARP2 in rod photoreceptors does cause structural alteration in older mice evidenced by a non-significant reduction in INL and IPL thickness and malformed outer segments as a function of age, resembling some of the previous observations in GARP2 overexpression mice (Chakraborty et al., 2015; DeRamus et al., 2017).

### Functional changes observed in GARP2-KO ERG responses

To investigate the physiology of the transgenic retina, we first performed a functional investigation using ERG recordings. To examine photoreceptor and rod bipolar cell function in the GARP2-KO, we performed scotopic

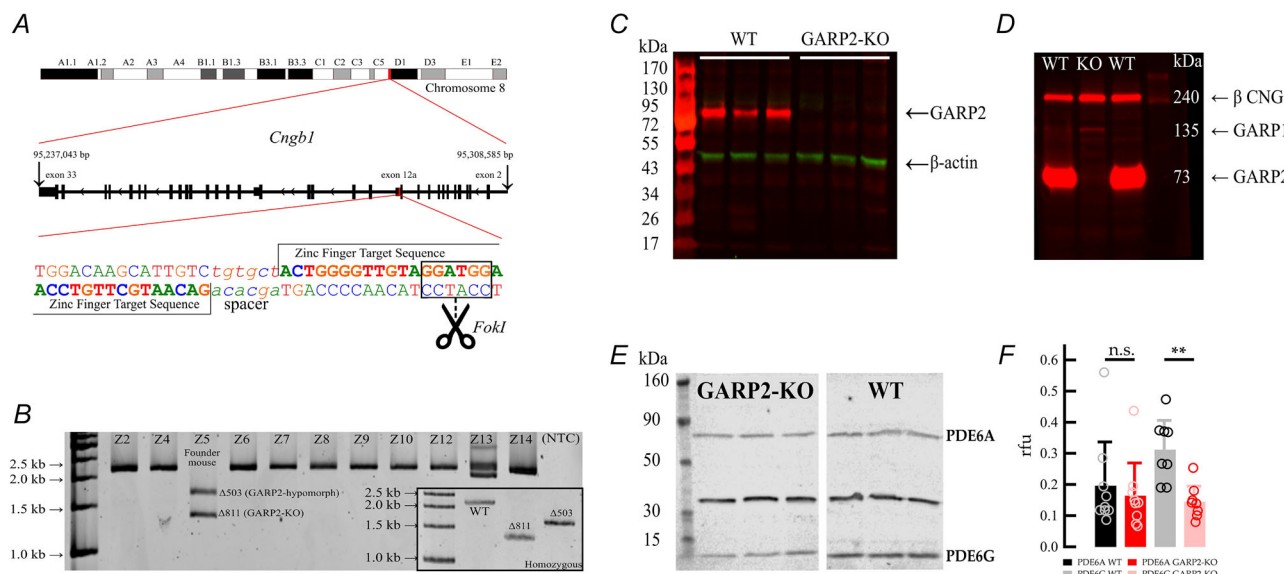

**Figure 1. Specific deletion of GARP2 in rod photoreceptors by ZFN-mediated gene editing**
*A*, *Cngb1* is located on the q arm of murine chromosome 8 and is composed of 33 exons. The full 33-exon transcript encodes the β-subunit of the CNG channel that is vital to the phototransduction cascade. GARP2 is encoded through an alternative splicing event at exon 12a. This exon encodes the last 8 amino acids of GARP2 and is only used to generate GARP2, not affecting the expression of CNGB1 or GARP1. A zinc finger nuclease created to bind specifically to a sequence near the GARP2-specific exon 12a was used to make a double-strand break, which, when repaired by non-homologous end joining, can lead to a targeted deletion. *B*, PCR genotyping performed on the ZFN mRNA-injected litter containing the founder mouse. Two deletions occurred in Z5, the founder mouse: Δ503 bp is missing the 3′ untranslated region and splice acceptor site, and Δ811 bp is missing all of the GARP2 specific exon 12a. Each deletion was bred to homozygosity (inset). The Δ503 deletion was a hypomorphic expressor of GARP2, and the longer Δ811 deletion is a complete knockout of GARP2 and the focus of this study. *C*, western blot analysis of retinal homogenates from 3 WT and 3 GARP2-KO mice. We used a GARP2-specific antibody and a β-actin antibody as control protein loading. GARP2 was not detected in the knockout mice. *D*, western blot analysis was performed on total retinal homogenates using a common N-terminal antibody to assess the presence of GARPs and the β-subunit in PM7 WT and homozygous GARP2-KO mice retinas. In WT and mutant mice samples, the 240 kDa β-subunit and the more abundant 73 kDa GARP2 are apparent. GARP1 (135 kDa) is difficult to discern in WT mice with this antibody but is apparent in the knockout, which may indicate some overexpression. GARP2 is not detectable in the mutant retina. *E*, representative immunoblots for PDE6A and PDE6G (β-actin for normalization) in WT and GARP2-KO mouse retinas. Single bands of appropriate size were detected for all products. *F*, expression levels of PDE6A were comparable between WT and GARP2-KO retinas (WT: 0.197 ± 0.140, $n = 8$ retinas; GARP2-KO: 0.164 ± 0.104, $n = 9$ retinas; $P = 0.6081$), whereas PDE6G is significantly reduced in the GARP2-KO (WT: 0.312 ± 0.094, $n = 8$ retinas; GARP2-KO: 0.145 ± 0.05, $n = 8$ retinas; $P = 0.002$). Relative fluorescent units (rfu) were determined by densitometry, normalized to β-actin levels and graphed.

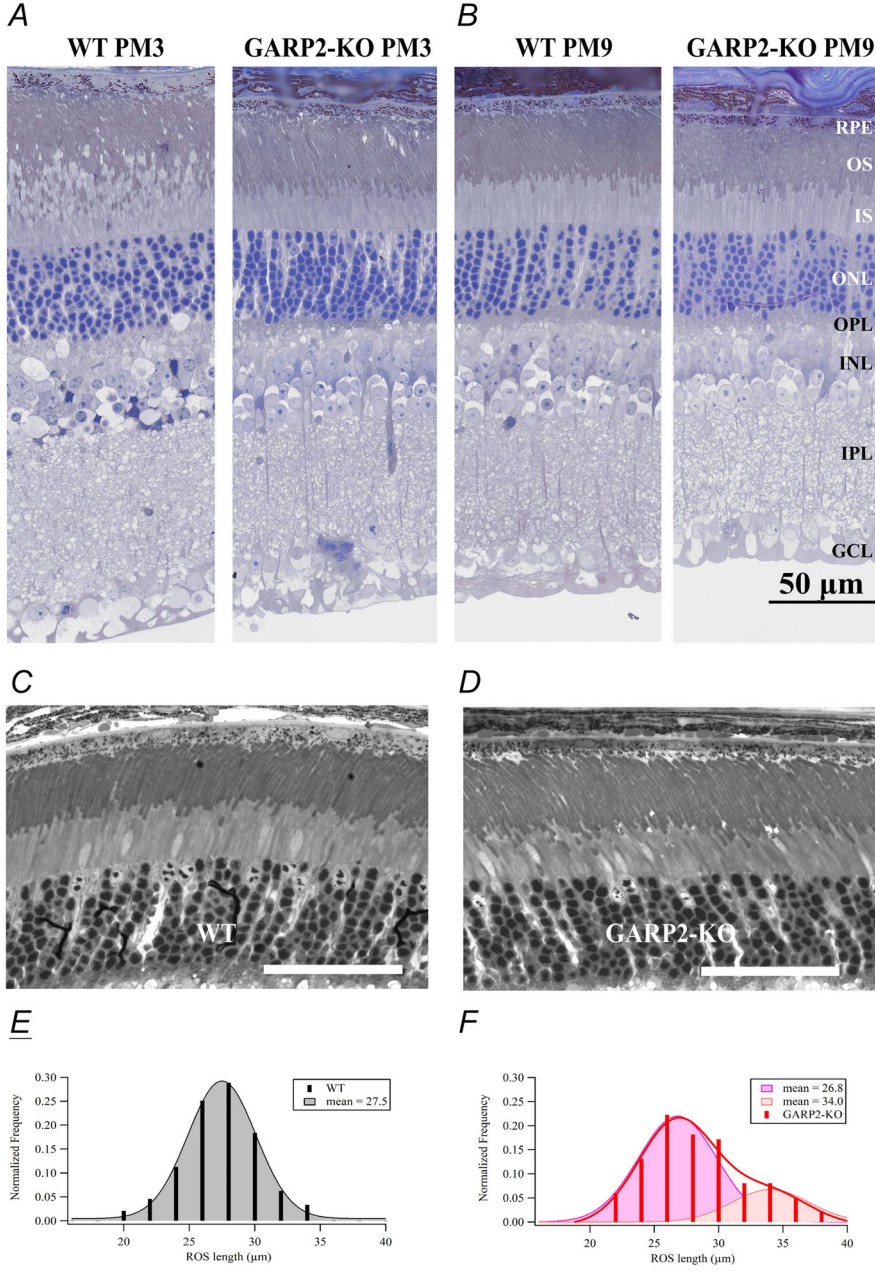

**Figure 2. Morphology of the GARP2-KO retina shows minor structural changes compared to WT**

*A*, Toluidine blue-stained LM sections of PM3 WT and GARP2-KO retinas show no significant structural changes (INL: WT, 49.3 ± 1.5 μm, *n* = 5 retinas; GARP2-KO, 49.7 ± 1.3 μm, *n* = 5 retinas; *P* = 0.68; IPL: WT, 65.7 ± 2.3 μm, *n* = 5 retinas; GARP2-KO, 58.8 ± 1.6 μm, *n* = 5 retinas; *P* = 0.1). *B*, Toluidine blue-stained LM sections of PM9 WT and GARP2-KO retinas show insignificant structural changes between WT and GARP2-KO with a strong trend in reduction of inner layer retinal thickness (INL: WT, 49.5 ±1.7 μm, *n* = 5 retinas; GARP2-KO, 42.5 ± 1.8 μm, *n* = 5 retinas; *P* = 0.13; IPL: WT, 62.1 ± 2. μm, *n* = 5 retinas; GARP2-KO, 52.7 ± 1.8 μm, *n* = 5 retinas; *P* = 0.06). *C* and *D*, Toluidine blue-stained sections of WT and GARP2-KO retina, respectively. Scale bars = 50 μm. ROS lengths vary across the GARP2-KO retina, with an average increased length compared to WT of about 1 μm (3.75% increase, WT, 25.1 ± 2.67 μm, *n* = 207 rods; GARP2-KO, 26 ± 5.1 μm, *n* = 133 rods; *P* = 0.03). *E*, histogram of WT ROS length by the frequency of occurrence normalized by the percentage of total ROS lengths counted, fit with a Gaussian distribution (peak at 27.5 ± 0.08 μm). *F*, a multipeak Gaussian curve fit of the GARP2-KO distribution of lengths reveals a biphasic distribution with peaks at 26.8 ± 0.8 μm (full width at half maximum (FWHM) 7.2 ± 1.4, height: 0.22 ± 0.03, total fit area: 1.7 ± 0.5) and 34.0 ± 2.6 μm (FWHM: 7.0 ± 4.1, Area: 0.51 ± 0.4, height: 0.07 ± 0.03; total fit area: 2.18 ± 0.63), with 23% of the ROS lengths belonging to a group with a mean longer than WT and 77% in a group with a mean shorter than WT. Scale bars = 50 μm. GCL: ganglion cell layer; INL: inner nuclear layer; IPL: inner plexiform layer; IS: inner segment; ONL: outer nuclear layer; OPL: outer plexiform layer; OS: outer segment; RPE: retinal pigment epithelium.

**Table 1. Comparison of observed morphological features of WT and GARP2-KO retinas**

|  | WT PM3 | GARP2-KO PM3 | WT PM9 | GARP2-KO PM9 |
|---|---|---|---|---|
| OS thickness (μm) | 33.5 ± 1.1 (5) | 33.6 ± 1.2 (5) | 33.6 ± 1.1 (5) | 34.6 ± 1.3 (5) |
| IS thickness (μm) | 23.4 ± 1.1 (5) | 20.3 ± 0.5 (5) | 20.9 ± 1.1 (5) | 20.8 ± 0.7 (5) |
| ONL thickness (μm) | 46.5 ± 1.1 (5) | 46.3 ± 1 (5) | 42.6 ± 0.8 (5) | 40.3± 0.9 (5) |
| INL thickness (μm) | 49.3 ± 1.5 (5) | 49.7 ± 1.3 (5) | 49.5 ±1.7 (5) | 42.5 ± 1.8 (5) |
| IPL thickness (μm) | 65.7 ± 2.3 (5) | 58.8 ± 1.6 (5) | 62.1 ± 2.6 (5) | 52.7 ± 1.8 (5) |
| GCL thickness (μm) | 25.4 ± 0.5 (5) | 27.1 ± 1.4 (5) | 28.7 ± 4.8 (5) | 24.6 ± 1.3 (5) |

Morphological measurements of WT and GARP2-KO mouse retinas at PM3, PM6 and PM9. Values are shown as means ± SD and number of retinas (*n*). GCL, ganglion cell layer; INL, inner nuclear layer; IPL, inner plexiform layer; IS, inner segment; ONL, outer nuclear layer; OS, outer segment.

ERGs at PM3. Figure 3*A* shows representative ERG traces at PM3 for the GARP2-KO animals compared to age-matched WT. At this stage, no change in a-wave or b-wave amplitude was observed between the genotypes (Fig. 3*B*,Table 2; WT *n* = 6 mice; GARP2-KO *n* = 7 mice), indicating no major functional deficits in the absence of GARP2. Since GARP2 is also a part of the CNG-channel β-subunit at its N-terminus, several β-subunit knockout mice are being studied as models for

RP. In this study, we wanted to determine if the selective deletion of soluble GARP2 exhibited any change in rod photoreceptor function with age. Thus, we next performed scotopic ERGs at PM6 and PM9. There was no change in a-wave or b-wave amplitude at PM6 (Fig. 3*C*, Table 2; WT *n* = 4 mice, GARP2-KO *n* = 5 mice), but interestingly a reduction of both a-wave and b-wave maximum amplitudes in GARP2 animals was observed at PM9 (Fig. 3*D*). The photoreceptor a-wave response amplitudes

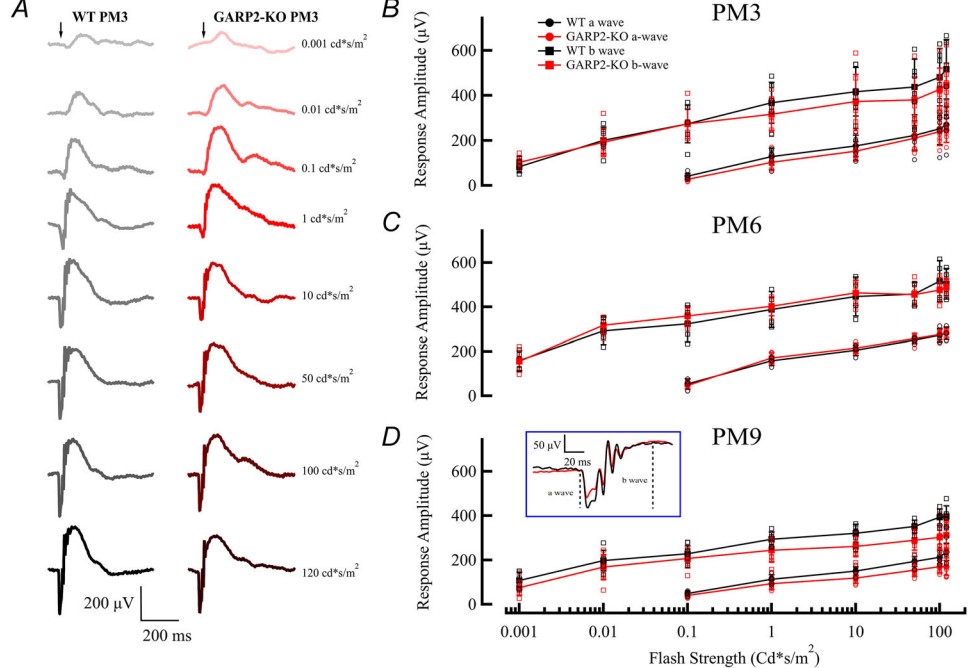

**Figure 3. Old GARP2-KO mice have attenuated a- and b-wave ERG response amplitudes**
*A*, average ERG response traces of WT and GARP2-KO at PM3 to flashes of increasing light intensities. *B*, intensity–response plot of average ERG a- (circles) and b-waves (squares) for WT (black) and GARP2-KO (red) rods at PM3 (WT *n* = six mice; GARP2-KO *n* = 7 mice; a-wave: *P* = 0.11; b-wave: *P* = 0.069). *C*, intensity–response plot of average ERG a- (circles) and b-waves (squares) for WT (black) and GARP2-KO (red) rods at PM6 (WT *n* = 4 mice; GARP2-KO *n* = 5 mice; a-wave: *P* = 0.466; b-wave: *P* = 0.789). *D*, intensity–response plot of average ERG a- (circles) and b-waves (squares) for WT (black) and GARP2-KO (red) rods at PM9 (WT *n* = six mice, GARP2-KO *n* = 9 mice; a-wave: $P = 3.55 \times 10^{-8}$; b-wave: $P = 1.62 \times 10^{-5}$).

**Table 2. Comparison of *in vivo* ERG responses in WT and GARP2-KO mice**

| | WT | | | GARP2-KO | | |
|---|---|---|---|---|---|---|
| | PM3 | PM6 | PM9 | PM3 | PM6 | PM9 |
| $I_{0.5}$ a wave (cd s/m$^2$) | 1.7 ± 0.9 (6) | 1.1 ± 0.4 (4) | 1.5 ± 0.2 (6) | 2.0 ± 0.9 (7) | 1 ± 0.2 (5) | 1.5 ± 0.12 (9) |
| $V_{max}$ a-wave (µV) | 270 ± 78 (6) | 282 ± 28 (4) | 236 ± 38 (6) | 246 ± 54 (7) | 284 ± 20 (5) | 170 ± 39 (9) *** |
| $I_{0.5}$ b-wave (cd s/m$^2$) | 0.03 ± 0.01 (6) | 0.01 ± 0.005 (4) | 0.03 ± 0.01 (6) | 0.03 ± 0.03 (7) | 0.01 ± 0.001 (5) | 0.03 ± 0.01 (9) |
| $V_{max}$ b-wave (µV) | 516 ± 132 (6) | 505 ± 68 (4) | 396 ± 48 (6) | 447 ± 96 (7) | 488 ± 38 (5) | 313 ± 63 (9) *** |

Recordings of ERG responses from WT and GARP2-KO mice to 20 ms 505 LED flashes at PM3, PM6 and PM9. Values are shown as means ± SD and number of mice recorded (*n*). $I_{0.5}$, half-maximal flash strength; $V_{max}$, maximal response amplitude.

**Table 3. Response properties of rods**

| | WT rods | | | GARP2-KO rods | | |
|---|---|---|---|---|---|---|
| | PM3 | PM6 | PM9 | PM3 | PM6 | PM9 |
| $I_{0.5}$ (R*/rod) | 20.8 ± 6.3 (17) | 21.4 ± 3.4 (12) | 24 ± 7.8 (13) | 16.5 ± 4.3 (8) | 21.7 ± 9.6 (16) | 27.3 ± 6.6 (20) |
| $I_{max}$ (pA) | 23.4 ± 6.2 (17) | 24.8 ± 2.5 (12) | 27 ± 7.4 (12) | 26.8 ± 10.2 (8) | 22.3 ± 7.3 (16) | 24.7 ± 7.4 (20) |
| $n$ | 1.3 ± 0.2 (17) | 1.4 ± 0.3 (12) | 1.3 ± 0.4 (13) | 1.3 ± 0.1 (8) | 1.4 ± 0.2 (16) | 1.1 ± 0.2 (20) |
| $T_{peak}$ (ms) | 143 ± 17 (17) | 144 ± 32 (12) | 135 ± 11 (13) | 132 ± 16 (8) | 136 ± 25 (16) | 129 ± 13 (20) |
| $V_m$ (mV) | −34.6 ± 7 (13) | −36.2 ± 5 (9) | −35 ± 7 (13) | −32.2 ± 6 (6) | −35.9 ± 8 (9) | −37.6 ± 5 (18) |
| $C$ (pF) | 3.6 ± 0.7 (15) | 3.5 ± 1.1 (11) | 4.3 ± 1.1 (13) | 3.4 ± 1.2 (7) | 3.8 ± 1.1 (12) | 4.5 ± 1 (19) |
| Noise (pA$^2$) | 0.17 ± 0.07 (7) | 0.2 ± 0.09 (6) | 0.18 ± 0.08 (8) | 0.07 ± 0.02 (4) * | 0.09 ± 0.05 (7) * | 0.17 ± 0.07 (8) |

Collected data of physiological properties of rod photoreceptors in WT and GARP2-KO mice at PM3, PM6 and PM9. Values are shown as means ± SD and number of cells (*n*). The resting membrane potential was recorded after light stimulations. $I_{0.5}$, half-maximal flash strength; $I_{max}$, maximal response amplitude; $n$, Hill coefficient; $T_{peak}$, time-to-peak; $V_m$, resting membrane potential; $C$, capacitance.

showed a significant reduction in the GARP2-KO mice, suggesting the overall light-evoked photocurrent was reduced in the retinas of GARP2-KO mice at this age. We also observed a significant reduction in the response amplitude of the b-wave, arising in downstream circuitry (Fig. 3*D*, Table 2; WT *n* = 6 WT mice, GARP2-KO *n* = 9 mice). To further assess if the observed changes had any effect on sensitivity and synaptic transmission in the rods, we calculated light sensitivity from the best-fit Hill curve to normalized light-intensity relations (Fig. 4). There was no significant change in the light sensitivity ($I_{1/2}$) in either a- or b-waves between the genotypes over time (Fig. 4*A–C*, Table 2; WT *n* = 6 mice, GARP2-KO *n* = 9 mice). Thus, the reduction in total response amplitude at PM9 could result from compromised photocurrent in the malformed rod outer segments observed (Fig. 2*D* and *F*), a minor reduction in the total number of rod photoreceptors, but with no major effects on light-activated phototransduction or synaptic transmission from rods to second-order RBCs.

## Functional changes observed in GARP2-KO single-cell responses

The full-field ERG response is a complex signal from the whole retina, so we next wanted to more thoroughly investigate the physiology of individual cells in the retinal circuitry. We first looked at the functionality of the photoreceptors by recording light-evoked responses from dark-adapted rods in mice at PM3, PM6 and PM9 (Fig. 5*A–C*). Interestingly, the functional properties of rod photoreceptors at these ages showed no significant differences from WT (Fig. 5*D*, Table 3; PM3 WT *n* = 17 rods, four mice; GARP2-KO *n* = 8 rods, three mice; PM6 WT *n* = 12 rods; six mice, GARP2-KO *n* = 17 rods six mice; PM9 WT *n* = 13 rods, four mice; GARP2-KO rods *n* = 20, six mice). The amplitude, sensitivity, time to peak and membrane voltage of rods showed no statistical differences and were comparable to previous responses seen in the literature (Beier et al., 2022; Pahlberg et al., 2017). Previously, phototransduction gain was found to be increased in a GARP2-Ox model (Sarfare et al.,

2014), with a >3-fold increase in the GARP2 expression combined with a reduction in OS length. This stems from greatly increased encounters of PDE6 with GARP2, with an expected increase in gain and $T_{\text{peak}}$. Surprisingly, our experiments looking at the kinetics of the light response revealed no significant differences between the gain (Fig. 6A–C; WT $n = 11$ rods, six mice; GARP2-KO $n = 11$ rods, seven mice) or recovery time constant (Fig. 6D; WT $n = 20$ rods, six mice; GARP2-KO $n = 17$ rods, seven mice) in our GARP2-KO rods compared to WT at PM6. We hypothesize that the absence of GARP2 does not have a major effect on the diffusional encounter of light-activated PDE6–GARP2, with minimal effect on altering photo-transduction gain. Taken together, all our observations indicate that the interaction of GARP2 with PDE6 may affect the structural integrity of the retina over time (see Fig. 2) but has a minor role in light-driven scotopic photo-transduction in rod photoreceptors.

We next looked at downstream signalling by recording light-evoked responses from dark-adapted RBCs in mice at PM3, 6 and 9 (Fig. 7A–C). As for the rods, the amplitude, sensitivity, time to peak and membrane voltage of RBCs showed no statistical differences (Fig. 7D, Table 4; WT PM3 $n = 30$ RBCs, two mice; GARP2-KO $n = 30$ RBCs, five mice; WT PM6 $n = 30$ RBCs, five mice; GARP2-KO $n = 25$ RBCs five mice; WT PM9 $n = 29$ RBCs, six mice; GARP2-KO $n = 35$ RBCs, seven mice), and were comparable to previous responses seen in the literature (Beier et al., 2022; Pahlberg et al., 2017). This suggests that most synapses between rods and RBCs targeted by our

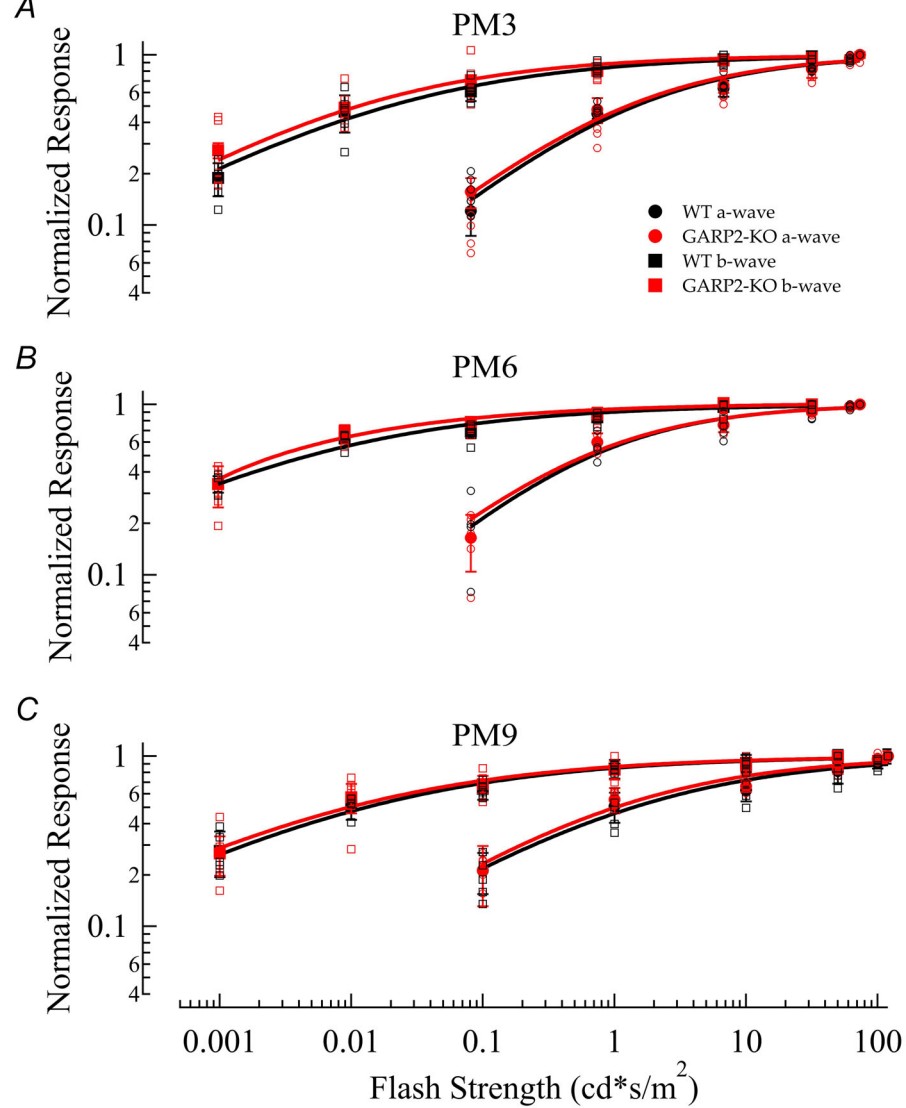

**Figure 4. Sensitivity of the light response is unchanged in isolated GARP2-KO retinas**
A–C, intensity–response plot (*n* as in Fig. 3 for all PMs; a-wave *P* = 0.764, b-wave *P* = 0.57) for ERG a- (circles) and b-waves (squares) of WT (black) and GARP2-KO (red) rods at PM3, PM6 and PM9 (see Table 2 for details).

**Table 4. Response properties of RBCs**

| | WT RBCs | | | GARP2-KO RBCs | | |
|---|---|---|---|---|---|---|
| | PM3 | PM6 | PM9 | PM3 | PM6 | PM9 |
| $I_{0.5}$ (R*/rod) | 3.5 ± 0.7 (30) | 2.9 ± 0.8 (30) | 3.6 ± 0.9 (29) | 3.3 ± 1 (30) | 2.9 ± 0.8 (25) | 3.7 ± 1.2 (35) |
| $I_{max}$ (pA) | 325 ± 116 (30) | 292 ± 154 (30) | 254 ± 115 (29) | 300 ± 104 (30) | 246 ± 140 (25) | 218 ± 105 (35) |
| $n_H$ | 1.7 ± 0.2 (29) | 1.9 ± 0.2 (30) | 1.7 ± 0.2 (29) | 1.6 ± 0.2 (30) | 1.9 ± 0.4 (25) | 1.8 ± 0.4 (35) |
| $T_{peak}$ (ms) | 105 ± 16 (30) | 102 ± 24 (30) | 109 ± 16 (29) | 103 ± 13 (30) | 99 ± 17 (24) | 113 ± 24 (35) |
| $V_m$ (mV) | −38 ± 5 (28) | −39 ± 7 (17) | −40 ± 7 (28) | −39 ± 7 (29) | −38 ± 7 (15) | −38 ± 6 (34) |
| $C$ (pF) | 5.6 ± 0.7 (29) | 6.5 ± 0.7 (26) | 5.9 ± 0.9 (25) | 6.1 ± 1 (28) | 7 ± 1 (19) | 6.3 ± 1 (32) |

Collected data of physiological properties of RBCs from PM3, 6 and 9 WT and GARP2-KO mice. Values are shown as means ± SD and number of cells (*n*). The resting membrane potential was recorded after light stimulations. $I_{0.5}$, half-maximal flash strength; $I_{max}$, maximal response amplitude; $n_H$, Hill coefficient; $T_{peak}$, time-to-peak; $V_m$, resting membrane potential; $C$, capacitance.

recordings, function normally at these time points, and the interaction of GARP2 with PDE6 on phototransduction has a minor role in synaptic transmission at the synaptic level between rods and RBCs.

Since we observed a downregulation of the inhibitory subunit PDE6G (Fig. 2), and GARP2 was reported to interact with PDE6 and regulate its basal activity in darkness *in vitro* (Pentia et al., 2006), we next compared the cellular dark noise in WT and GARP2-KO rod

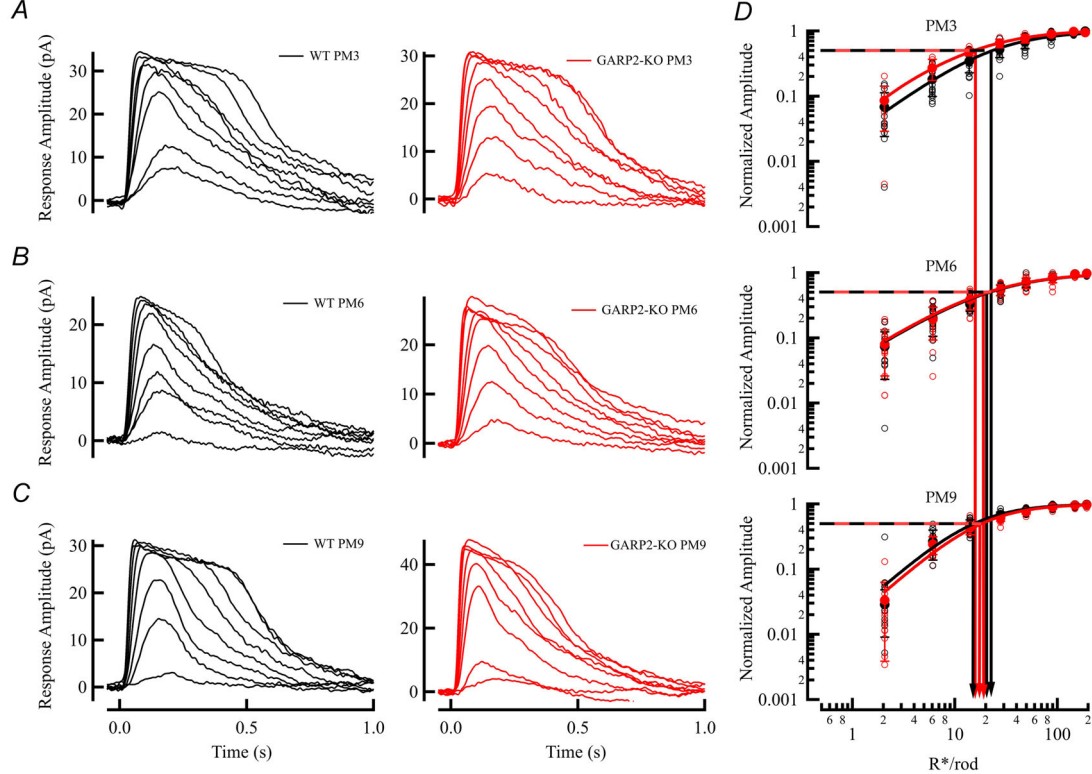

**Figure 5. Physiological light response properties of single GARP2 rod photoreceptors are similar to WT**
*A*, representative light-evoked responses from PM3 WT (black, holding membrane potential −40 mV) and GARP2-KO rods (red), respectively. Light stimuli delivered at 0 s. *B*, representative light-evoked responses from 6M WT (black) and GARP2-KO rods (red), respectively. *C*, representative light-evoked responses from 9M WT control (black) and GARP2-KO rods (red), respectively. *D*, intensity–response plot of WT (black) and GARP2-KO (red) rods at PM3, PM6 and PM9. The sensitivity ($I_{1/2}$) was derived from the best-fit Hill equation (PM3 WT *n* = 17 rods, four mice; GARP2-KO *n* = 8 rods, three mice; PM6 WT *n* = 12 rods, six mice; GARP2-KO *n* = 17 rods, six mice; PM9 WT *n* = 13 rods, four mice; GARP2-KO *n* = 20 rods, six mice. Maximal response amplitude ($I_{max}$) *P* = 0.729, sensitivity ($I_{0.5}$), *P* = 0.699, time to peak ($T_{peak}$), *P* = 0.748, and membrane voltage ($V_m$) *P* = 0.276).

photoreceptors (Fig. 8*A–C*, Table 3). The two dominant forms of cellular dark noise interfering with photon detection in rods have been demonstrated to arise from spontaneous activation of the visual pigment and basal PDE6 activity (Baylor et al., 1980; Rieke, & Baylor, 1996). Although a recent study on a transgenic mouse with a genetically modified rhodopsin surprisingly and controversially suggest visual pigment could play a role in continuous noise (Chai et al., 2024), the majority of evidence in the field has shown that continuous noise originates from the downstream transduction cascade and more specifically PDE6 (Ashtakova et al., 2017; Baylor et al., 1980, 1979; Bocchero & Pahlberg, 2023; Lamb, 1987; Reingruber et al., 2015; Rieke & Baylor, 1996). Thus, if the lack of GARP2 in the GARP2-KO mice of our study increases the basal activity of PDE6 in darkness, it could reduce the fluctuations in cGMP concentration, which will be seen as a reduction in continuous noise (Morshedian et al., 2022; Reingruber et al., 2015, 2020). Excitingly, the continuous noise level between 0.6 and 10 Hz showed a significant reduction in GARP2-KO mice compared to WT at both PM3 and 6 (Fig. 8*D* and *E*, Table 3; PM3: WT $n = 7$ rods, three mice; GARP2-KO $n = 4$ rods, two mice; PM6: WT $n = 6$ rods, four mice; GARP2-KO $n =$

7 rods, four mice). We observed a ∼two-fold reduction in the total cumulative power of the noise in the absence of GARP2, indicating a role for GARP2 in controlling the basal activity of PDE, and thus cGMP turnover rate (Fig. 8). Interestingly, at PM9 the dark noise levels were similar between the genotypes (PM9: WT $n =$ eight rods, four mice; GARP2-KO $n =$ eight rods, four mice), indicating an actual increase in noise levels over time in GARP2-KO mice, again suggesting a role for GARP2 in maintaining the functionality and physiology of rod photoreceptors over time.

## Discussion

In this study, we have successfully generated a selective knockout of soluble GARP2 in the rod photoreceptors of the mouse retina and observed both structural and functional changes that deviate from WT rods. ROS were transiently longer than WT within certain regions of the GARP2-KO retina. The overall thicknesses of the GARP2-KO retinal layers were comparable through PM9; however, a minor reduction in inner retinal thickness was observed. The ERG responses showed amplitude reductions over time, with a significant reduction of the

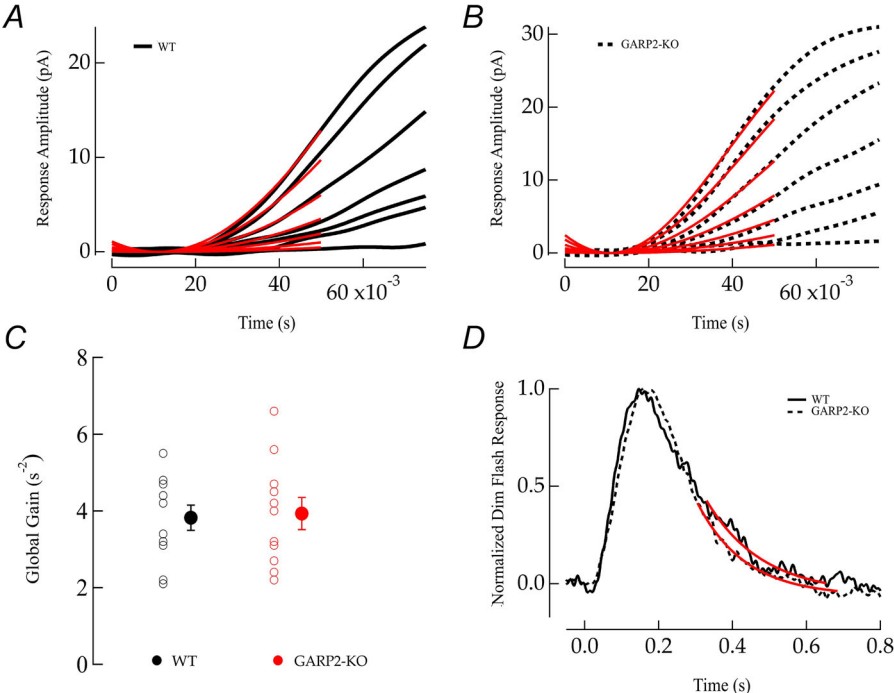

**Figure 6. No significant alterations in phototransduction gain and response recovery in GARP2-KO rods**
*A*, ensemble fit (red traces) of the Lamb–Pugh model of phototransduction gain in WT rods. *B*, ensemble fit (red traces) of the Lamb–Pugh model of phototransduction gain in GARP2-KO rods. *C*, average global gain in WT and GARP2-KO rods (*A* (s$^{-2}$): WT 3.8 ± 1, $n = 11$ rods, six mice; GARP2-KO 3.9 ± 1.3, $n = 11$ rods, seven mice; $P = 0.689$). *D*, average dim flash responses from WT control (continuous) and GARP2-KO rods (dashed), respectively. Recovery time constant ($\tau_{rec}$) was estimated from the exponential fit (red traces) to the final 35% slope of the average dim flash response ($\tau$ (ms): WT 251 ± 242, $n = 20$ rods, six mice; GARP2-KO 233 ± 200, $n = 17$ rods, seven mice; $P = 0.81$).

photoreceptor-driven photocurrent in GARP2 mice at PM9. Surprisingly, the functional properties of individual rod photoreceptors and rod bipolar cells showed no significant differences, indicating normal function at the rod-to-rod bipolar synapse, leaving a possible role for GARP2 in maintaining structural integrity and health of rods as a function of age. Our most compelling finding is a striking change in the basal activity of photoreceptors, seen as a significant decrease in continuous dark noise, suggesting an important role for GARP2 in controlling the structural stability and basal activity of PDE6 to regulate this critical aspect of photoreceptor function.

### Structural changes in GARP2-KO mice

We have previously shown that knockout of the β-subunit and GARPs leads to retinal degeneration (*Cngb1*-X1 KO) (Zhang et al., 2009), and knockout of only the β-subunit without effecting soluble GARP expression (*Cngb1*-X26 KO) (Hüttl et al., 2005) leads to a retinal degeneration with features that are distinct from the X1-KO (X26 mice have almost no detectable channel α-subunit and corresponding near total loss of phototransduction with rapid complete loss of photoreceptors). Due to the phenotypic differences, we hypothesized that selective GARP2 knockout would alter the structure and function of the rod photoreceptor in a manner distinct from the phenotypes observed in the X1 or X26 knockouts. *In vivo* and *ex vivo* observations of the GARP2-KO mouse showed GARP2 to be histologically very similar to WT (cf. Fig. 2*A*) early on, but with some cell loss in the inner retina by 9 months (cf. Fig. 2*B*). Moreover, ROS are indeed longer than WT within certain regions of the GARP2-KO retina (Fig. 2). These changes are not always observed, and the regions where they appear are not constricted to any quadrant or discernible location (i.e. more central or peripheral, near the optic nerve, near the ora serrata, etc.). The

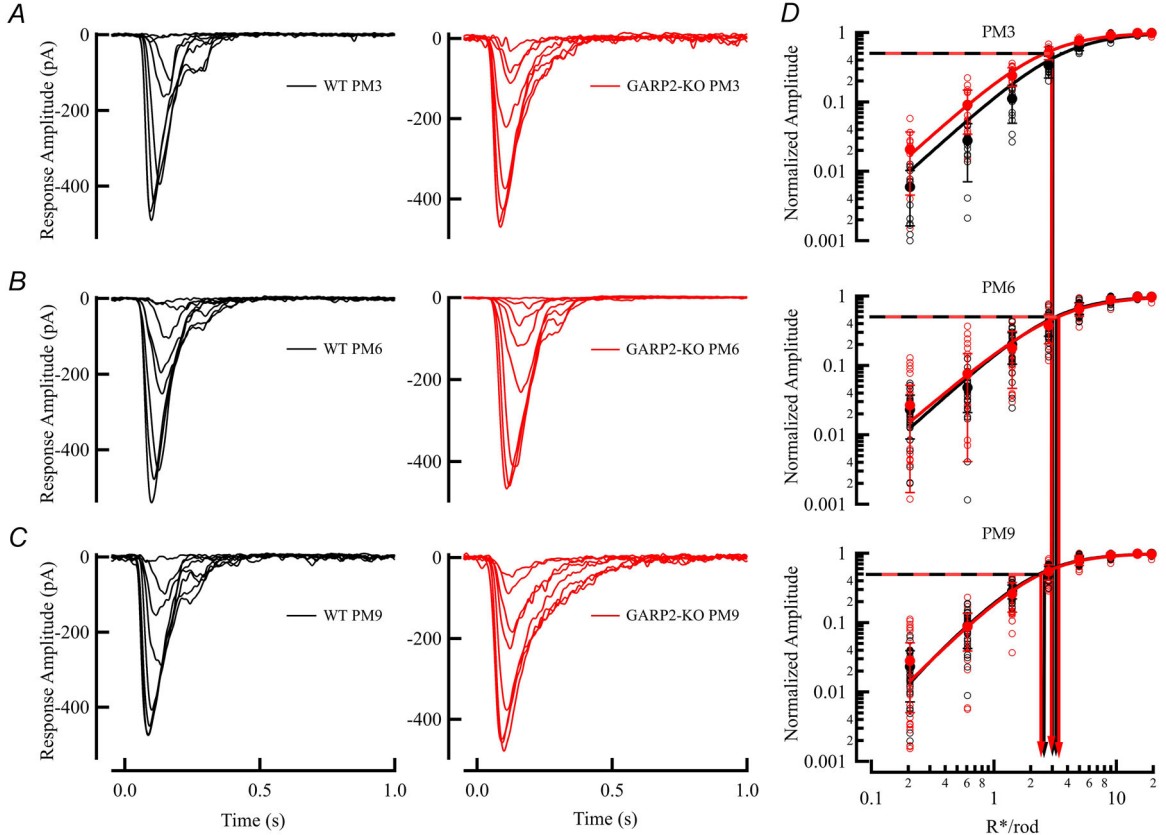

**Figure 7. Physiological light response properties of individual GARP2 rod bipolar cells are unchanged compared to WT**
*A*, representative light-evoked responses from PM3 WT (light grey, holding membrane potential -60mV) and GARP2-KO (pink) RBCs, respectively. *B*, representative light-evoked responses from PM6 WT (grey) and GARP2-KO rods (red), respectively. *C*, representative light-evoked responses from PM9 WT control (black) and GARP2-KO rods (brown), respectively. *D*, intensity–response plot of WT and GARP2-KO RBCs (flash family in *A–C* are representative of recordings across many cells. Flash delivered at 0 s (PM3 WT *n* = 30 RBCs, two mice; GARP2-KO *n* = 30 RBCs five mice; PM6 WT *n* = 30 RBCs, five mice; GARP2-KO *n* = 25 RBCs, five mice; PM9 WT *n* = 29 RBCs, six mice; GARP2-KO *n* = 35 RBCs, seven mice. $I_{max}$ *P* = 0.0625, $I_{1/2}$ *P* = 0.78, $T_{peak}$ *P* = 0.748, and $V_m$, *P* = 0.959).

basis for this may be due to incomplete compensation for the absence of GARP2 by GARP1. GARP1 and GARP2 are identical in sequence for the first 318 amino acids, with GARP2 ending with eight residues encoded by an alternative exon. GARP1 extends another 232 amino acids, and the last 16 are unique to GARP1. So, the extra sequence in GARP1 may reduce the binding affinity for GARP2 targets. Additionally, since GARP1 is about 20-fold less abundant than GARP2 it may not be present in sufficient amounts to fully compensate for GARP2 function. Although not an indication of lasting retinal pathology, the appearance of the overgrown ROS indicates at least a minor role for GARP2 in stabilizing the disc membranes and preventing retinal degeneration. Thus, while the absence of GARP2 leads to changes in physio-logical responses over time, the PDE6–GARP2 inter-action could also stabilize this complex to disc membranes and thus prevent disorganization of the discs and retinal degeneration with age.

### Decline in the light-driven photo response of the retina over time

Functionally, by ERG analyses, the GARP2-KO mouse was not different from WT up to PM6. However, at PM9, the GARP2-KO mouse exhibited deficits in the functional properties analysed, as both scotopic a-wave and b-wave amplitudes were reduced (Fig. 3, Table 2). Analysis of both the GARP2-KO a- and b-wave light sensitivities indicated, that although the response amplitude is decreased, the

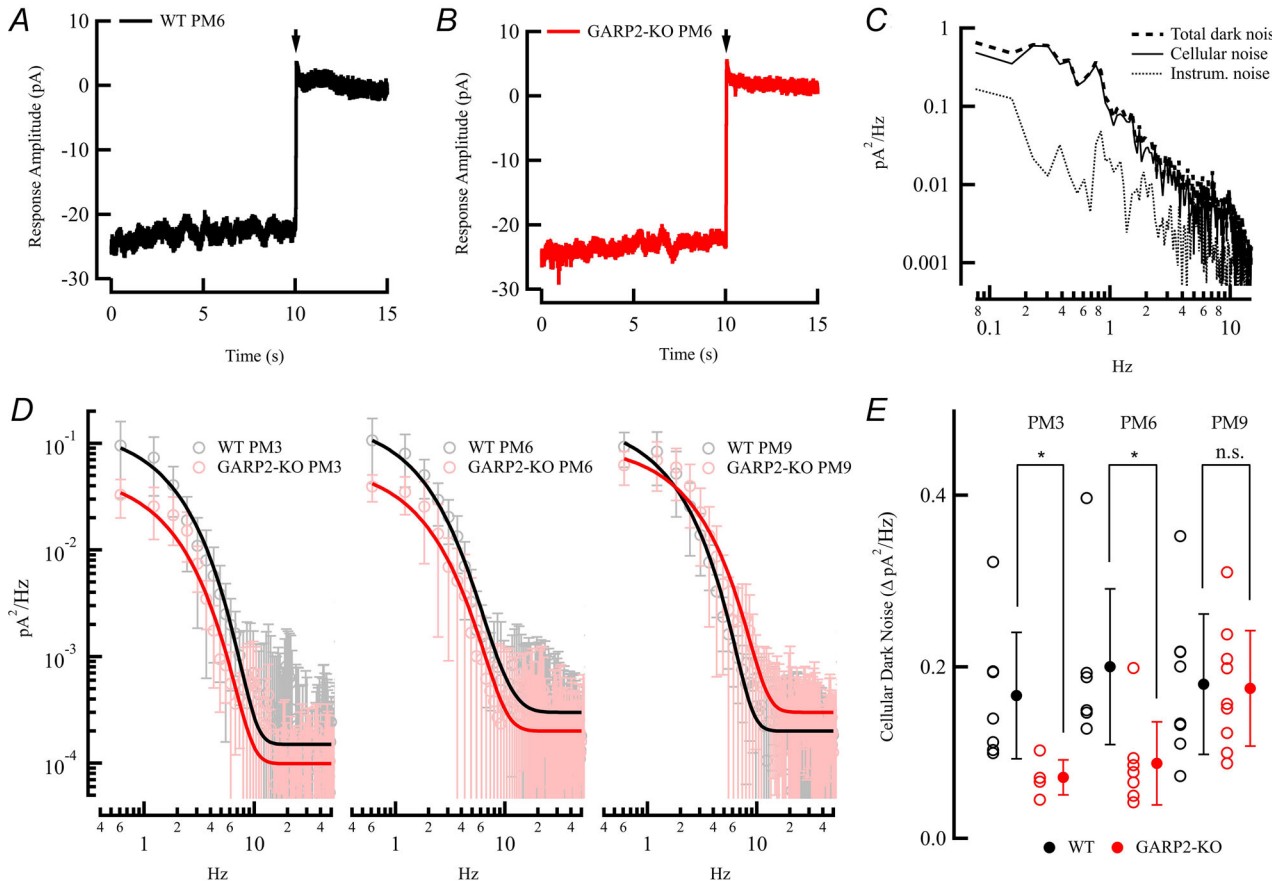

**Figure 8. Continuous receptor noise in rods in complete darkness is significantly altered in the absence of GARP2**

*A*, representative recording of membrane currents from a WT rod (black). Whole-cell patch-clamp recordings were done in complete darkness, followed by a saturating light flash (≥ 200 R*/rod) to close all the CNG channels. *B*, representative recording of membrane currents from a GARP2-KO rod (black), as in *A*. *C*, continuous noise power spectra from a representative WT rod (black). Spectra were constructed by fast Fourier transformation of the membrane current of total dark noise (dashed), instrumental noise (dotted), and cellular dark noise (continuous), derived by subtracting the instrumental noise (saturated response) from the total dark noise recordings. *D*, average cellular dark noise power spectra from WT (black) and GARP2-KO (red) rods. *E*, cumulative power of dark noise between 0.6 and 10 Hz from WT (black traces) and GARP2-KO (red traces) rods (PM3: WT *n* = 7 rods, three mice; GARP2-KO *n* = 4 rods, two mice, *P* = 0.026; PM6: WT *n* = 6 rods, four mice; GARP2-KO *n* = 7 rods, four mice *P* = 0.0095; PM9: WT *n* = 8 rods, four mice; GARP2-KO *n* = 8 rods, four mice, *P* = 0.97).

sensitivity is similar for both GARP2-KO and WT photoreceptors (Fig. 4, Table 2). This correlates with the occurrence of overgrown and malformed outer segments at PM9, which could explain a reduction in overall response amplitude by a decrease in number of cells responding to the light stimulus, or an increase in extracellular resistance of the recordings, with minor effects on overall light sensitivity from the majority of unaffected rods. Accordingly, our single-cell recordings showed no difference in rod or rod bipolar-cell physiological properties up to PM9, indicating normal function of rod phototransduction and most of the rod synapses in the absence of GARP2. It is possible that the single-cell recordings only measured signals from rods and bipolar cells with normal synaptic transmission, as these cells represent a majority of the cells. Overall, the limited number of major functional changes in GARP2-KO rod photoreceptors, combined with no sustained major structural deficits to retinal morphology up to PM6, is unexpected. To understand the basis for this, the decline in the ERG response amplitude and possible downstream circuitry defects observed in older mice will need further investigation.

## Significant alterations in the basal dark activity of photoreceptors

A key feature of the capability of rods to detect single photons (Baylor et al., 1979) is the optimization of molecular properties of proteins in the phototransduction cascade. Rods must produce a single photon response that rises above the intrinsic fluctuations produced by discrete and continuous noise events, big enough to be extracted by the downstream circuitry (Baylor et al., 1979; Field et al., 2019; Pahlberg, & Sampath, 2011). Hence, regulating the concentration and the basal spontaneous activity of phototransduction proteins is critical in setting the signal-to-noise ratio of rods. Accordingly, this study's most exciting and novel discovery was the profound change in continuous dark noise in the absence of GARP2, with no major significant alteration in other physiological properties of rods up to PM6.

It was previously shown that PDE6G, the inhibitory subunit of PDE6, was prevented from spontaneously dissociating from PDE6 in the presence of GARP2, with a significant effect on the basal dark activity of PDE6 (Pentia et al., 2006). Moreover, it has been shown that the continuous dark noise in rods is controlled by basal PDE6 activity, and a reduction in PDE6 concentration (and basal PDE6 activity) results in an increase in dark noise (Bocchero & Pahlberg, 2023; Rieke & Baylor, 1996). As shown in Fig. 1, the absence of GARP2 reduced the expression levels of the inhibitory PDE6G subunit, with no overall change in PDE6A (nor presumably in PDE6B) abundance. The downregulation of the inhibitory PDE6G subunit could be expected to increase basal PDE6 activity altering dark noise levels. This effect on basal PDE6 activity in darkness would then further exacerbate the increase in basal PDE6 activity the absence of GARP2 would be expected to have. Importantly for this study, the combined absence of GARP2 and reduction in PDE6G will lead to increased basal PDE6 activity, lowering fluctuations in the dark turnover rate of cGMP, which decreases fluctuations in CNG channel open probability, seen as a change in continuous noise (Fig. 8*E*) (Morshedian et al., 2022; Reingruber et al., 2015, 2020). The response sensitivity of the transduction cascade and kinetics of the light response are linked to the basal activity of PDE6 and its effect on cGMP turnover. However, published evidence has shown that the changes in PDE6 expression and basal activity needs to be significantly altered before the changes can be observed (Morshedian et al., 2022). Consequently, even reducing expression of PDE6B to ∼70% has little effect on sensitivity and kinetics but effects the level of continuous noise in rods (Bocchero et al., 2023). Interestingly, the aforementioned reduction in PDE6B leads to an increase in dark noise, the exact opposite of the effect shown in the current study. Taken together, the regulation of basal PDE6 activity and cGMP turnover rate within some acceptable limit, seems to have profound effects in the control of physiological properties of the light response. Our result suggests an important role for GARP2 in regulating the structure and basal activity of PDE6 in darkness.

If the absence of GARP2 decreases dark noise, why is it expressed? It has been shown, that an increase in basal PDE6 activity and faster dark turnover of cGMP will affect the amplification of the single photon response and possibly lead to faster rise times and more rapid response decay, leading to reduced light sensitivity (Reingruber et al., 2020). On the other hand, a significant reduction in the basal activity of PDE6 will increase the rod's single-photon response and sensitivity but also substantially increase dark noise (Abtout et al., 2021; Bocchero, & Pahlberg, 2023). Furthermore, a reduction in PDE6 enzyme concentration leads to non-uniformity in spontaneous PDE6 activity, greater variability in the dark-resting concentration of cGMP, and an increase in response variance and less reproducible responses (Morshedian et al., 2022). We hypothesize that by stabilizing the basal PDE6 activity, GARP2 could allow tuning the cGMP turnover rate without affecting PDE6 concentration and its distribution on the disc membrane. Thus, by reducing the basal activity of PDE6 without changes in concentration, GARP2 would allow the rods to both optimize PDE6 levels and maintain an optimal cGMP turnover rate to control dark noise and efficient transduction gain and shut off, for optimal single-photon detection.

Also, ATP consumption in rods is among the highest of any neurons, particularly in the dark state, which could have metabolic consequences for the health of the cells (Hurley et al., 2015; Okawa et al., 2008). An increase in PDE6 activity would decrease noise but increase the cGMP turnover rate. A higher basal cGMP turnover rate from a substantial increase in basal PD6E activity would thus inevitably lead to an increase in ATP consumption, which could be detrimental to the health of the rods (Okawa et al., 2008; Reingruber et al., 2013). By stabilizing basal PDE6 activity, GARP2 would counteract such effects and reduce possible ATP consumption without reducing overall PDE6 concentration and reliability of photon detection (Morshedian et al., 2022). Moreover, during light adaptation the basal PDE6 activity and cGMP concentration are altered from the dark resting state and the single photon regime, and the role of GARP2 for light and dark adaptation will require further studies.

In conclusion, although the GARP2–PDE6G interactions seem to increase the noise level (within apparently acceptable limits) in WT rods, cytoplasmic GARP2 may be required to maintain the structure and health of outer segments, stabilize the dark basal PDE6 activity and optimize the concentration of the PDE6 subunits on the disc membrane to control the size and duration of the single photon response. In other words, our results suggest that the regulation of basal PDE6 activity by GARP2 has a physiological role in setting the cGMP concentration at an appropriate level for optimal phototransduction, signal detection and signal-to-noise ratio in rod photoreceptors.

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

## Additional information

### Data availability statement

All data that support the findings of this study are available from the corresponding authors upon reasonable request.

### Competing interests

The authors declare no competing interests.

### Author contributions

D.A.S., U.B., M.L.D., M.N.N., T.W.K., S.J.P. and J.P. designed research; D.A.S., U.B., M.L.D., M.N.N., J.M., S.J.P. and J.P. performed research; U.B., T.W.K., M.N.N., S.J.P. and J.P. analysed data; D.A.S., U.B., M.L.D., M.N.N., T.W.K., S.J.P. and J.P. edited the paper.; D.A.S., U.B., M.N.N., S.J.P. and J.P. wrote the paper. All authors have read and approved the final version of this manuscript and agree to be accountable for all aspects of the work in ensuring that questions related to the accuracy or integrity of any part of the work are appropriately investigated and resolved. All persons designated as authors qualify for authorship, and all those who qualify for authorship are listed.

### Funding

This work was supported in part by the Intramural Research Programs of the National Institutes of Health Grants (J.P.), NIH grant EY018143 (S.J.P.), and an NEI Core Grant for vision research, P30 EY03039 (S.J.P.). The contributions of the NIH authors are considered Works of the United States Government. The findings and conclusions presented in this paper are those of the authors and do not necessarily reflect the views of the NIH or the U.S. Department of Health and Human Services.

### Acknowledgements

We thank Hua Tian of the Jeff Diamond lab for animal care and genotyping, and Gordon Fain for critical comments and suggestions on the manuscript.

### Keywords

cGMP turnover, dark noise, GARP2, PDE6, single photon detection

## Supporting information

Additional supporting information can be found online in the Supporting Information section at the end of the HTML view of the article. Supporting information files available:

**Peer Review History**

