## [Peer Review History · The Journal of Physiology]

Selective Knockout of Murine Glutamic Acid-rich Protein 2 (GARP2) Significantly Alters Dark Continuous Noise in Rod Photoreceptors

Delores A Stacks, Ulisse Bocchero, Marci L DeRamus, Mai N Nguyen, Jeffrey Messinger, Timothy W. Kraft, Steven Jay Pittler, and Johan Pahlberg

DOI: 10.1113/JP286548

Corresponding author(s): Johan Pahlberg (johan.pahlberg@nih.gov)

The following individual(s) involved in review of this submission have agreed to reveal their identity: Maureen A McCall (Referee #1)

Review Timeline:

Submission Date:	08-Apr-2024
Editorial Decision:	03-Jun-2024
Revision Received:	30-May-2025
Editorial Decision:	26-Aug-2025
Revision Received:	06-Oct-2025
Accepted:	29-Oct-2025

Senior Editor: Katalin Toth

Reviewing Editor: Karin Dedek

Transaction Report:

Dear Dr Pahlberg,

Re: JP-RP-2024-286548 "Selective Knockout of Murine Glutamic Acid-rich Protein 2 (GARP2) Significantly Alters Cellular Dark Noise in Rod Photoreceptors" by Delores A Stacks, Ulisse Bocchero, Marci L DeRamus, Mai N Nguyen, Jeffrey Messinger, Timothy W. Kraft, Steven Jay Pittler, and Johan Pahlberg

Thank you for submitting your manuscript to The Journal of Physiology. It has been assessed by a Reviewing Editor and by 2 Referees and relevant feedback from their reports is copied below for your information. We regret to inform you that your manuscript is not considered acceptable for publication in The Journal of Physiology.

However, as the reviewers highlighted a number of strengths in your manuscript, we would like to suggest that you consider submitting your manuscript to our sister journal, Experimental Physiology.

Experimental Physiology is a peer-reviewed, Open Access journal covering all areas of physiology, with particular emphasis on translation and integration.

Further details can be found here: [https://onlinelibrary.wiley.com/journal/10.1111/\(ISSN\)1469-445X](https://onlinelibrary.wiley.com/journal/10.1111/(ISSN)1469-445X).

You can find information on ways to pay Open Access Article Processing Charges (APCs), including a list of eligible consortia and institutions covered by Transformational Agreements with our publisher Wiley here: <https://authorservices.wiley.com/author-resources/Journal-Authors/open-access/affiliation-policies-payments/institutional-funder-payments.html>.

In 2023, over 80% of manuscripts transferred to Experimental Physiology went on to receive an acceptance decision after review.

If you would like to proceed with transferring your manuscript for consideration by the Editorial Board of Experimental Physiology, please do so through the following link: Link Not Available.

We would encourage you to take into account the review comments before completing your submission to Experimental Physiology. You are welcome to make any further changes you wish, and to replace the manuscript file with an updated one.

Yours sincerely,

Katalin Toth
Senior Editor
The Journal of Physiology

Reviewing Editor's comments:

Dear authors,

Thank you for submitting your work to the Journal of Physiology. Your manuscript was evaluated by two independent reviewers and me. While all reviewers agreed that the technical aspects of your work are very strong, they also raised substantial concerns regarding for example the statistical data evaluation (low n for some experiments, statistical tests for others, correction for multiple comparisons missing or not reported when t-tests were used), which also does not comply fully with journal standards (data reported as mean +- SEM not SD). In addition, one reviewer raised concerns about the discrepancy between the morphological findings (phenotype in 9-month-old mice) and the physiological findings (no phenotype in 9-months-old mice but in 3- and 6-months-old ones). Moreover, as the rationale for the study requires unaltered expression of the beta-subunit of the CNG channel (the other splice product apart from GARP2), the authors should consider demonstrating the actual expression levels.

Referee #1:

JP-RP-2024-286548

Selective Knockout of Murine Glutamic Acid-rich Protein 2 (GARP2) Significantly Alters Cellular Dark Noise in Rod Photoreceptors.

This is a clearly written manuscript describing the authors attempts to find a phenotype in this GARP2 KO mouse. A classis structural and number of function techniques have been brought to bear in order to determine if a phenotype exists and at what level.

Unfortunately, very little changes in these KO mice and the changes that the authors highlight in most cases are not backed up by a rigorous experimental design or a detailed description of the statistics that they used. As the manuscript stands this is significant work to be done before the differences that they say they observe in the abstract, introduction, results and methods are assured to the reviewer.

Details follow for the methods and results as there are many things to clarify before comments can be made on the introduction and discussion.

Methods:

1. Animal care. Are the WT and the KOs on the same background, e.g., C57Bl6J? saying that they are backcrossed for 5 generations does not mean that they share the same background strain (line 121).
2. What measurements were made of the layers the description is not sufficient and the n of numbers of eye (3) is too low for any valid statistical analysis (see table).
3. Were any of the animals in the ERG analysis used in the morphological analysis. NOT noted.
4. Were any of the animals in the ERGs at 3 months also analysed at later time points? NOT noted.
5. How did the authors confirm that they were recording from rod bipolar, where the cell filled and morphology examined? Did all bipolar cells have both an ascending and descending process, e.g., was their morphology complete?
6. The authors used a 2 way anova on the ERG data and multiple t tests on other data sets. The majority of the data is comparing across genotype and age, why didn't they use a 2 way ANOVA for those data sets?
7. In the GARP Kos, PDE6G is downregulated. I do not see how this is addressed as a confounding factor especially for the single rod data. If it is mentioned it is so important that more emphasis should be made.

Results - Morphology - an n of 3 is not sufficient to compute a mean and not even for an SD, although it can be done mathematically. None of the dubious p values is at or below 0.05. This means that none of the comparisons are different and the authors should not try to say things like "just below the level of significance" A comparison is either significant or not.

ERG function - stats appear appropriate. Although the overall F should be reported and then the individual comparisons are then reported. This should be corrected.

The sensitivity measures depend on the amplitude of the response and the max amplitude of the response, how can the ERGs at 9 month be different and the sensitivity is not. Can the authors better describe this?

The authors state that the reduction in total photocurrent at PM9 could result from a minor reduction in the total number of rod photoreceptors, or a reduction in OS length with less CNG channels, but with no major effects on light-activated phototransduction or synaptic transmission from rods to second-order RBCs. However none of the morphological differences are statically significant. This is a disconnect that must be addressed.

See above for questions regarding the identity of the single cells that were recorded.

The authors state "taken together Indicate that the interaction ... may affect the structural integrity of the retina over time. But again there is no evidence for a statistical difference in morphology.

Finally, although there is a significant difference in continuous noise at 3 and 6 months this does not occur at 9 months, which adds to the confusion in the data set presented. And more questions are raised than are addressed in the manuscript.

Each of these concerns needs to be addressed.

Discussion.

Eliminate the discussion of non significant effects.

Referee #2:

Stacks et al JP-RP-2024-286548

General:

This is a solid study that advances our understanding of the cellular mechanisms that limit vision in dim light levels, the range where neural noise interferes with visual signals. Two important sources of noise - discrete and continuous - originate in the rod photoreceptors and while the source of discrete noise is well understood, the same cannot be said about continuous noise where the molecular mechanisms remain undetermined. Here Stacks and colleagues present original data that provide new insights into the role that PDE and GARP2 play in the generation of continuous noise. This study builds on previous findings by both Dr Pahlberg and Dr Pittler. To test their hypotheses Dr Stacks et al generated and fully characterized a new genetic mouse model where GARP2 is specifically targeted and knocked out. Standard (but very challenging) electrophysiological tests at the retinal and single cell level determined a slow age-related degeneration in these mice. The major finding that the GARP2 KO has reduced continuous noise has important implications to the field, however, as discussed by the authors, the mechanistic interpretation of the results is slightly blurred given the associated changes in expression of PDE6G, the inhibitory subunit of the PDE complex. Despite this drawback, this study provides a solid step towards the understanding of noise and vision. The statistical tests and the number of replications seem appropriate for this kind of studies. The figures and graphs are well done and easy to read. I have a few comments listed below that may help improve some aspects of the manuscript.

Comments:

- The rationale for targeting exon 12a in the *cngb1* gene should be made explicit to help the readers understand the innovation brought by this new model and how it differs from the other existing mouse genetic models that target GARP, GARP2 and/or CNGB1.
- GARP2 is generated by alternative splicing of the *cngb1* gene. Is there any chance that knocking out exon12a could modify the expression sequence of the NT terminal and/or the expression levels of CNGB1? Changes in the NT could result in abnormal binding to PER thereby leading to degeneration. They could also modify the open/ closing kinetics of the CNG channel. Controls showing the sequence and expression levels of CNGB1 in the GARP2 knockout mouse can address these questions Do the authors have some of this data available to include in the manuscript?
- Same applies to GARP1.
- The single cell recordings are compelling, but it remains unclear why the postulated changes in PDE basal activity in the KO are not accompanied by changes in circulating currents in the dark, the resting potentials or even in recovery kinetics. Please comment on the failure of GARP2 KO to modulate these phototransduction properties because as far as I understand these three response properties are linked to 'steady' PDE activity.
- Related to the previous point. What are the expected changes in the level of basal PDE activity in the KO that result in the reduction of the noise levels? This information may provide a better sense of the observed physiology. The hypothesized

changes are backed largely by the biochemistry experiments of Pentic et al but not directly measured in the rods. Given that IBMX jump experiments are not possible in mammalian rods, could modeling, perhaps with the Reingruber model provide further insights?

Minor:

Structural changes in GARP2-KO. Please indicate what $n = 3$ represents. Is this number of sections or number of eyes, number of mice? How many replications? Are they taken from similar regions of retina? What is their distance from optic nerve?

Figure 4. Why are linear scales used in Figure 3 and log scales in Figure 4? Please justify.

Line 340. The ERGs at the more intense flashes may also activate cone bipolar cells via secondary or tertiary paths or even directly by cone activation. Please edit.

Table 3. WT rods at PM3 are 10-fold noisier than WT rods at PM6 and PM9. This difference does not seem to be statistically significant (not starred). It is unclear why it does not seem to reflect in the power spectra of the young PM3 rods. Please comment on the significance.

Table 3. This is a silly notation point. Maximal response in pA should be I_{max} , not V_{max} .

Fig. 2D. X-axis units should be $cd\ s/m^2$ not cd/m^2 . Please edit.

END OF COMMENTS

JP-RP-2024-286548

Selective Knockout of Murine Glutamic Acid-rich Protein 2 (GARP2) Significantly Alters Cellular Dark Noise in Rod Photoreceptors

This is a clearly written manuscript describing the authors attempts to find a phenotype in this GARP2 KO mouse. A classis structural and number of function techniques have been brought to bear in order to determine if a phenotype exists and at what level.

Unfortunately, very little changes in these KO mice and the changes that the authors highlight in most cases are not backed up by a rigorous experimental design or a detailed description of the statistics that they used. As the manuscript stands this is significant work to be done before the differences that they say they observe in the abstract, introduction, results and methods are assured to the reviewer.

Details follow for the methods and results as there are many things to clarify before comments can be made on the introduction and discussion.

Methods:

1. Animal care. Are the WT and the KOs on the same background, e.g., C57Bl6J? saying that they are backcrossed for 5 generations does not mean that they share the same background strain. (line 121)
2. What measurements were made of the layers the description is not sufficient and the n of numbers of eye (3) is too low for any valid statistical analysis. (see table)
3. Were any of the animals in the ERG analysis used in the morphological analysis. NOT noted
4. Were any of the animals in the ERGs at 3 months also analysed at later time points? NOT noted.
5. How did the authors confirm that they were recording from rod bipolar, where the cell filled and morphology examined? Did all bipolar cells have both an ascending and descending process, e.g., was their morphology complete?
6. The authors used a 2 way anova on the ERG data and multiple t tests on other data sets. The majority of the data is comparing across genotype and age, why didn't they use a 2 way ANOVA for those data sets?
7. In the GARP Kos, PDE6G is downregulated. I do not see how this is addressed as a confounding factor especially for the single rod data. If it is mentioned it is so important that more emphasis should be made.

Results – Morphology – an n of 3 is not sufficient to compute a mean and not even for an SD, althous it can be done mathematically. None of the dubious p values is at or below 0.05. This means that none of the comparisons are different and the authors should not try to say things like “just below the level of significance” A comparison is either significant of not.

ERG function – stats appear appropriate. Although the overall F should be reported and then the individual comparisons are then reported. This should be corrected.

The sensitivity measures depend on the amplitude of the response and the max amplitude of the response, how can the ERGs at 9 month be different and the sensitivity is not. Can the authors better describe this?

The authors state that the reduction in total photocurrent at PM9 could result from a minor reduction in the total number of rod photoreceptors, or a reduction in OS length with less CNG channels, but with no major effects on light-activated phototransduction or synaptic transmission from rods to second-order RBCs. However none of the morphological differences are statically significant. This is a disconnect that must be addressed.

See above for questions regarding the identity of the single cells that were recorded.

The authors state “taken together Indicate that the interaction ... may affect the structural integrity of the retina over time. But again there is no evidence for a statistical difference in morphology.

Finally, although there is a significant difference in continuous noise at 3 and 6 months this does not occur at 9 months, which adds to the confusion in the data set presented. And more questions are raised than are addressed in the manuscript.

Each of these concerns needs to be addressed.

Discussion.

Eliminate the discussion of non significant effects.

Reviewing Editor's comments:

Dear authors,

Thank you for submitting your work to the Journal of Physiology. Your manuscript was evaluated by two independent reviewers and me.

While all reviewers agreed that the technical aspects of your work are very strong, they also raised substantial concerns regarding for example the statistical data evaluation (low n for some experiments, statistical tests for others, correction for multiple comparisons missing or not reported when t-tests were used), which also does not comply fully with journal standards (data reported as mean +- SEM not SD).

We thank the Reviewing Editor for pointing out the data evaluation brought up by Rev #1. Per Journal policy: “*The Journal of Physiology* supports Format Free initial submissions (FFIS). For initial submissions authors are not required to conform to journal style.” “Standard Deviation (SD) without '±' must be used instead of Standard Error of the Mean (SEM), unless the use of SEM is fully justified and reported alongside confidence intervals.” We have added confidence intervals as required by the Journal to all Figure Legends where appropriate, and performed new experiments to increase n. We are happy to further revise the manuscript to Journal style as the manuscript is accepted for publication.

In addition, one reviewer raised concerns about the discrepancy between the morphological findings (phenotype in 9-month-old mice) and the physiological findings (no phenotype in 9-months-old mice but in 3- and 6-months-old ones).

The physiological data mentioned here by the Reviewing Editor would apply to the difference in dark noise in the 3- and 6-M old mice. This is the **main finding of the paper** and the essence of the whole study, which is noted by Rev #2. The reduction in noise in the absence of GARP2 is a novel finding, and it helps push the field forward in understanding the origin of dark noise in rods, extensively studied for 5 decades, but still not fully understood. This reduction seems to diminish at 9-M, and “the lack of phenotype” is a *de facto* **increase** in noise in the **old** GARP2 mice, compared to 3 and 6 M. This coincides with our new data, showing overgrowth of rod outersegments in old GARP2 KO mice. The increase in dark noise is a common trait of rod degeneration or rods that will start to malform. Thus, this observation highlights the deficiency in function and possibly structure, with the lack of GARP2 in mice as they get older.

Moreover, as the rationale for the study requires unaltered expression of the beta-subunit of the CNG channel (the other splice product apart from GARP2), the authors should consider demonstrating the actual expression levels.

Apart from the transiently malformed rod outersegments demonstrated by our new set of experiments, the morphology and structure of the rods show no observations of significant difference (with the methods used) between the WT and the GARP2 transgenic mice. Knockout

of the beta-subunit with (Huttl et al. 2005, PMID:15634774) or without (Zhang et al, 2009, PMID:19339551) GARP expression leads to major changes in morphology and subsequent degeneration. Moreover, there is no significant difference in dark current as recorded by our single cell recordings at any stages between the genotypes, indicating normal expression of the CNG channel. Nevertheless, we have provided experimental evidence of the expression of the channel.

Referee #1:

JP-RP-2024-286548

Selective Knockout of Murine Glutamic Acid-rich Protein 2 (GARP2) Significantly Alters Cellular Dark Noise in Rod Photoreceptors.

This is a clearly written manuscript describing the authors attempts to find a phenotype in this GARP2 KO mouse. A classis structural and number of function techniques have been brought to bear in order to determine if a phenotype exists and at what level.

We thank the Reviewer for this kind comment on our manuscript.

Unfortunately, very little changes in these KO mice and the changes that the authors highlight in most cases are not backed up by a rigorous experimental design or a detailed description of the statistics that they used. As the manuscript stands this is significant work to be done before the differences that they say they observe in the abstract, introduction, results and methods are assured to the reviewer.

With these comments in mind the work has been improved to support our conclusions.

Details follow for the methods and results as there are many things to clarify before comments can be made on the introduction and discussion.

Methods:

1. Animal care. Are the WT and the KOs on the same background, e.g., C57Bl6J? saying that they are backcrossed for 5 generations does not mean that they share the same background strain (line 121).

The mouse lines are backcrossed with C57Bl6J for > 5 generations, by this point for several years (in fact the mice studied are at least congenic at this point). Backcrossing for >5 generations is common and accepted praxis in the field, to justify experiments as being done in the same genetic background.

2. What measurements were made of the layers the description is not sufficient and the n of

numbers of eye (3) is too low for any valid statistical analysis (see table).

We have added to the description and done more experiments to increase n, and now have significant statistical power to do valid analysis, albeit with the same outcome.

3. Were any of the animals in the ERG analysis used in the morphological analysis. NOT noted.

The morphological and structural expertise was provided by the Laboratory of Dr. Pittler at UAB in Birmingham, Alabama. The physiological expertise was provided by the Laboratory of Dr. Pahlberg at the National Institutes of Health in Bethesda, MD. The experiments were performed on the same mouse lines, as stated, bred at the Animal Facilities at NIH. This is common practice in the field in collaborative works done between different Institutes. For the *in vivo* ERG experiments some, but not all experiments, were done on the same mice as they were aging, and some mice were used for patch clamp and morphology studies as well. All the retinal tissue was used in every experiment, to maximize data collection and minimize the use of experimental animals. Most importantly, all experiments were performed on the exact same mouse in all cases. We have added a line to Methods to state this (Line 124).

4. Were any of the animals in the ERGs at 3 months also analyzed at later time points? NOT noted.

See above.

5. How did the authors confirm that they were recording from rod bipolar, where the cell filled and morphology examined? Did all bipolar cells have both an ascending and descending process, e.g., was their morphology complete?

The laboratory of Dr. Pahlberg is one of the leaders in the field of electrophysiological patch clamp recordings from the retina, with more than two decades of experience in such recordings. Rods constitute 97% of photoreceptors and ~40-50% of all cells in the INL are bipolar cells, and around half, or ~200k, are RBCs in C57/BL6 mouse (Jeon et al., 1998; Strettoi et al., 2010; Keeley et al., 2022). The shape of the rod and RBC somas, their position in the ONL and INL, respectively, the kinetics of the response and their amplitudes, are very stereotypical and classifying the cells is not an issue in the patch clamp recordings done in retinal slices. This approach has been used for decades in the field.

6. The authors used a 2 way anova on the ERG data and multiple t tests on other data sets. The majority of the data is comparing across genotype and age, why didn't they use a 2 way ANOVA for those data sets?

The 2-way ANOVA was used for ERG experiments, when comparing the whole intensity response curve at different ages. For patch clamp-recordings the comparison was always pairwise at the same age, and no data was used twice, which would justify the use of t-test. Nevertheless, we have performed ANOVA on all experiments to address any concerns.

7. In the GARP Kos, PDE6G is downregulated. I do not see how this is addressed as a

confounding factor especially for the single rod data. If it is mentioned it is so important that more emphasis should be made.

Downregulation of the inhibitory PDE6G subunit is expected to increase the basal PDE6 activity, which is the same effect as the absence or downregulation of GARP2. This is stated in the manuscript text, and we have now added a few words to emphasize this fact (Line 485).

Results - Morphology - an n of 3 is not sufficient to compute a mean and not even for an SD, although it can be done mathematically. None of the dubious p values is at or below 0.05. This means that none of the comparisons are different and the authors should not try to say things like "just below the level of significance" A comparison is either significant or not.

The n refers to the number of mice, however the data was collected from several slides in each animal, averaged and then used for statistical analysis. We have done more experiments to increase n and have significant statistical power to validate our analyses. As the Reviewer correctly noted, we have removed any reference to "just below the level of significance" where no statistical significance was found, and we appreciate the correction.

ERG function - stats appear appropriate. Although the overall F should be reported and then the individual comparisons are then reported. This should be corrected.

The sensitivity measures depend on the amplitude of the response and the max amplitude of the response, how can the ERGs at 9 month be different and the sensitivity is not. Can the authors better describe this?

We respectfully disagree with the reviewer. The sensitivity measure is completely different from the comparison of max amplitude and does NOT immediately depend on the amount of dark current. The amount of dark current in ERG recordings depends on the total amount of retina used in the recordings, number of photoreceptors or the average length of the rod outer segments. This is not directly associated with the efficiency of the phototransduction cascade of individual rods or RBCs to amplify the absorption of a photon and signal the detection of light. Hence, the half-maximal response is taken as a measure of the sensitivity of the rods or RBCs, which, by definition, is independent of the maximum current. E.g., for single cell recordings in WT mice, the total dark current may differ depending on pipette and input resistance, the speed of break in, the length of the recording, and several other factors. However, on average, RBCs with 200 pA or 600 pA responses will have indistinguishable $I_{1/2}$ (or sensitivity) regardless. Thus, a change in ERG currents could reflect a subtle change in rod or retinal structure, that could not be discerned by the light microscopy experiments done, but transiently shown by Electron Microscopy. The transduction machinery seems to be unaffected, as shown by similar sensitivities. This difference adds to the notion of differences, structurally and functionally, at older age (>9M) and the significance of the study. We have added some language to clarify this (Line 360).

The authors state that the reduction in total photocurrent at PM9 could result from a minor reduction in the total number of rod photoreceptors, or a reduction in OS length with less CNG channels, but with no major effects on light-activated phototransduction or synaptic transmission

from rods to second-order RBCs. However, none of the morphological differences are statically significant. This is a disconnect that must be addressed.

See above.

See above for questions regarding the identity of the single cells that were recorded.

See above answer for point 5.

The authors state "taken together Indicate that the interaction ... may affect the structural integrity of the retina over time. But again there is no evidence for a statistical difference in morphology.

We have added new experiments showing changes in morphology including overgrowth of outer segments and subsequent misalignment of disks. These changes were not appreciated until we performed electron microscopy to analyze structure. These new data support our conclusion that the lack of GARP2 leads to structural changes that will affect the overall ERG response of the whole retina.

Finally, although there is a significant difference in continuous noise at 3 and 6 months this does not occur at 9 months, which adds to the confusion in the data set presented. And more questions are raised than are addressed in the manuscript.

The reduction in noise in the absence of GARP2 is a novel finding, and it helps to guide the field forward in understanding the origin of dark noise in rods, extensively studied for 5 decades, but still not fully understood. This reduction seems to diminish at 9-M, and “the lack of phenotype” is actually **an increase** in noise in the **old** GARP2 mice, compared to 3- and 6-M. The increase in dark noise is a hallmark of rod degeneration or rods that will start to malform. Thus, this observation highlights the deficiency in function, and possibly structure, with the lack of GARP2 in mice, as they get older. We agree that the findings of our paper open several interesting questions, and it will provide our laboratories with several years of interesting work to further guide the field forward in understanding signal and noise properties of rods, and the degeneration and structure of the retina. They are obviously beyond the scope of this publication.

Each of these concerns needs to be addressed.

All concerns have been addressed to the best of our knowledge

Discussion.

Eliminate the discussion of non significant effects.

Discussion has been modified and any mention of non-significant results removed.

Referee #2:

Stacks et al JP-RP-2024-286548

General:

This is a **solid study** that **advances our understanding of the cellular mechanisms** that limit vision in dim light levels, the range where neural noise interferes with visual signals. Two important sources of noise - discrete and continuous - originate in the rod photoreceptors and while the source of discrete noise is well understood, the same cannot be said about continuous noise where the molecular mechanisms remain undetermined. Here Stacks and colleagues **present original data that provide new insights into the role that PDE and GARP2 play** in the generation of continuous noise.

This study **builds on previous findings by both Dr Pahlberg and Dr Pittler**. To test their hypotheses Dr Stacks et al generated and fully characterized a new genetic mouse model where GARP2 is specifically targeted and knocked out. Standard **(but very challenging)** electrophysiological tests at the retinal and single cell level determined a slow age-related degeneration in these mice. The major finding that the GARP2 KO has reduced continuous noise **has important implications to the field**, however, as discussed by the authors, the mechanistic interpretation of the results is slightly blurred given the associated changes in expression of PDE6G, the inhibitory subunit of the PDE complex. Despite this drawback, this study provides a **solid step towards the understanding of noise and vision**.

We thank the reviewer for these wonderful comments. We appreciate that the reviewer found the work to be an excellent contribution to the field. We agree that the findings will have important implications for the field and will advance our understanding of signal and noise properties in the retina and guide the field forward.

The **statistical tests and the number of replications seem appropriate** for this kind of studies. **The figures and graphs are well done and easy to read.**

We thank the reviewer for this kind notion.

I have a few comments listed below that may help improve some aspects of the manuscript.

Comments:

- The rationale for targeting exon 12a in the *cngb1* gene should be made explicit to help the readers understand the innovation brought by this new model and how it differs from the other existing mouse genetic models that target GARP, GARP2 and/or CNGB1.

We have added to the text in both the Material and Methods and the results sections to make the rational clear. Put simply, exon 12a encodes the last 8 amino acids of GARP2 and is unique to

GARP2. Thus, deletion of exon 12a selectively eliminates GARP2 expression without major effects on the expression of the beta-subunit and GARP1.

- GARP2 is generated by alternative splicing of the *cngb1* gene. Is there any chance that knocking out exon12a could modify the expression sequence of the NT terminal and/or the expression levels of CNGB1? Changes in the NT could result in abnormal binding to PER thereby leading to degeneration. They could also modify the open/ closing kinetics of the CNG channel. Controls showing the sequence and expression levels of CNGB1 in the GARP2 knockout mouse can address these questions Do the authors have some of this data available to include in the manuscript?

See the above response. Except for the occasional malformed rod outersegments at later stages, the morphology and structure of the rods show no difference between the WT and the GARP2 transgenic mice. Moreover, there is no significant difference in dark current at any stages between the genotypes, indicating normal expression of the CNG channel. Nevertheless, we have provided experimental evidence of the expression of the channel.

- The single cell recordings are compelling, but it remains unclear why the postulated changes in PDE basal activity in the KO are not accompanied by changes in circulating currents in the dark, the resting potentials or even in recovery kinetics. Please comment on the failure of GARP2 KO to modulate these phototransduction properties because as far as I understand these three response properties are linked to 'steady' PDE activity.

Indeed, as correctly noted by the reviewer, the response kinetics are linked to the basal activity of PDE. However, there is published evidence that the changes in PDE expression and basal activity needs to be significant before the changes occur (Morshedean et al., 2022). Also, reducing expression of PDE6 to almost half (Bocchero et al., 2023), has little effect on sensitivity and kinetics, but effects the level of continuous noise in rods. Thus, our manipulation of GARP2 seems to be sufficient to alter dark noise and possibly the health of the retina over time, whereas the effects on kinetics are minor. We have added some language to the Discussion to point out the issue brought up by the Reviewer (Line 493).

- Related to the previous point. What are the expected changes in the level of basal PDE activity in the KO that result in the reduction of the noise levels? This information may provide a better sense of the observed physiology. The hypothesized changes are backed largely by the biochemistry experiments of Pentia et al but not directly measured in the rods. Given that IBMX jump experiments are not possible in mammalian rods, could modeling, perhaps with the Reingruber model provide further insights?

We agree with the reviewer, the absolute changes in PDE6 activity would add to the results. However, these experiments are not trivial and very time consuming, and beyond the expertise of any of our labs. We strongly believe previous published work (Pentia et al., 2006), and our physiological results will suffice to make the proposed conclusions of the manuscript.

Minor:

Structural changes in GARP2-KO. Please indicate what $n = 3$ represents. Is this number of sections or number of eyes, number of mice? How many replications? Are they taken from similar regions of retina? What is their distance from optic nerve?

The n refers to the number of mice, however the data was collected from several sections in each animal, averaged and then used for statistical analysis. The sections were always taken from similar regions. We have analyzed more mice to increase n .

Figure 4. Why are linear scales used in Figure 3 and log scales in Figure 4? Please justify.

The changes in absolute amplitude of the ERG currents are shown on a linear scale, as is common for these kinds of recordings. This way the visualization of differences is clear to see for the reader. Fig 4. shows light sensitivity, which is reported as half maximal light intensity ($I_{1/2}$). Thus, sensitivity is not dependent on absolute amplitude, and the fitting of the Hill curve and the difference in sensitivity, or lack thereof, is clearer on a logarithmic scale, where the initial part of the fit to a flash family becomes a straight line.

Line 340. The ERGs at the more intense flashes may also activate cone bipolar cells via secondary or tertiary paths or even directly by cone activation. Please edit.

We thank the reviewer for this very accurate comment. As stated in the manuscript, the ERG b-wave is a measure of signaling downstream of rods, which is dominated by RBCs but also may have contributions from ON-cone BC and possibly other cell types. This is stated in the manuscript, as we do not claim to measure RBCs in these recordings but signaling downstream of photoreceptors. To carefully study RBC (and individual rod) responses, we are using single cell recordings.

Table 3. WT rods at PM3 are 10-fold noisier than WT rods at PM6 and PM9. This difference does not seem to be statistically significant (not starred). It is unclear why it does not seem to reflect in the power spectra of the young PM3 rods. Please comment on the significance.

We thank the reviewer for this critical observation. This is a mistake on our part that has occurred during analysis and re-analysis of the noise data and editing the manuscript. The values have now been corrected to match those of Fig 8.

Table 3. This is a silly notation point. Maximal response in pA should be I_{max} , not V_{max} .

This is not silly, but very accurate. It has now been corrected.

Fig. 2D. X-axis units should be $cd\ s/m^2$ not cd/m^2 . Please edit.

This has been corrected.

References:

- Bocchero U & Pahlberg J (2023). Origin of Discrete and Continuous Dark Noise in Rod Photoreceptors. *eNeuro* **10**, ENEURO.0390-23.2023.
- Hüttl S, Michalakis S, Seeliger M, Luo DG, Acar N, Geiger H, Hudl K, Mader R, Haverkamp S, Moser M, Pfeifer A, Gerstner A, Yau KW, Biel M (2005). Impaired channel targeting and retinal degeneration in mice lacking the cyclic nucleotide-gated channel subunit CNGB1. *J Neurosci.* **25**(1):130-8.
- Jeon CJ, Strettoi E & Masland RH (1998). The Major Cell Populations of the Mouse Retina. *J Neurosci* **18** (21) 8936-8946
- Keeley PW., Patel SS & Reese BE (2023). Cell numbers, cell ratios, and developmental plasticity in the rod pathway of the mouse retina. *J Anatomy*, **243**, 204–222
- Morshedian A, Sendek G, Ng SY, Boyd K, Radu RA, Liu M, Artemyev NO, Sampath AP & Fain GL (2022). Reproducibility of the Rod Photoreceptor Response Depends Critically on the Concentration of the Phosphodiesterase Effector Enzyme. *J Neurosci* **42**, 2180–218
- Pentia DC, Hosier S & Cote RH (2006). The glutamic acid-rich protein-2 (GARP2) is a high affinity rod photoreceptor phosphodiesterase (PDE6)-binding protein that modulates its catalytic properties. *J Biol Chem* **281**, 5500–5505
- Strettoi E, Novelli E, Mazzoni F, Barone I & Damiani D (2010). Complexity of retinal cone bipolar cells. *Prog Retin Eye Res.* **29**(4):272-83.
- Zhang Y, Molday LL, Molday RS, Sarfare SS, Woodruff ML, Fain GL, Kraft TW, Pittler SJ. Knockout of GARPs and the β -subunit of the rod cGMP-gated channel disrupts disk morphogenesis and rod outer segment structural integrity. *J Cell Sci.* **122**:1192-200.

Dear Dr Pahlberg,

Re: JP-RP-2025-286548R1-A "Selective Knockout of Murine Glutamic Acid-rich Protein 2 (GARP2) Significantly Alters Dark Continuous Noise in Rod Photoreceptors" by Delores A Stacks, Ulisse Bocchero, Marci L DeRamus, Mai N Nguyen, Jeffrey Messinger, Timothy W. Kraft, Steven Jay Pittler, and Johan Pahlberg

Thank you for submitting your manuscript to The Journal of Physiology. It has been assessed by a Reviewing Editor and by 1 expert referee and we are pleased to tell you that it is acceptable for publication following satisfactory revision.

REVISION CHECKLIST:

Please upload two versions of your manuscript text: one with all relevant changes highlighted and one clean version with no changes tracked. The manuscript file should include all tables and figure legends, but each figure/graph should be uploaded as separate, high-resolution files. The journal is now integrated with Wiley's Image Checking service. For further details, see: <https://www.wiley.com/en-us/network/publishing/research-publishing/trending-stories/upholding-image-integrity-wileys->

image-screening-service

We look forward to receiving your revised submission.

Yours sincerely,

Katalin Toth
Senior Editor
The Journal of Physiology

REQUIRED ITEMS

- You must upload original, uncropped western blot/gel images (including controls) if they are not included in the manuscript. This is to confirm that no inappropriate, unethical or misleading image manipulation has occurred. These should be uploaded as 'Supporting information for review process only'. Please label/highlight the original gels so that we can clearly see which sections/lanes have been used in the manuscript figures. For more information, see: <https://physoc.onlinelibrary.wiley.com/hub/journal-policies#imagmanip>.

- Papers must comply with the Statistics Policy: https://jp.msubmit.net/cgi-bin/main.plex?form_type=display_requirements#statistics.

In summary:

- If $n \leq 30$, all data points must be plotted in the figure in a way that reveals their range and distribution. A bar graph with data points overlaid, a box and whisker plot or a violin plot (preferably with data points included) are acceptable formats.

- If $n > 30$, then the entire raw dataset must be made available either as supporting information, or hosted on a not-for-profit repository, e.g. FigShare, with access details provided in the manuscript.

- 'n' clearly defined (e.g. x cells from y slices in z animals) in the Methods. Authors should be mindful of pseudoreplication.

- All relevant 'n' values must be clearly stated in the main text, figures and tables.

- The most appropriate summary statistic (e.g. mean or median and standard deviation) must be used. Standard Error of the Mean (SEM) alone is not permitted.

- Exact p values must be stated. Authors must not use 'greater than' or 'less than'. Exact p values must be stated to three significant figures even when 'no statistical significance' is claimed.

- A Data Availability Statement is required for all papers reporting original data. This must be in the Additional Information section of the manuscript itself. It must have the paragraph heading 'Data Availability Statement'. All data supporting the results in the paper must be either: in the paper itself; uploaded as Supporting Information for Online Publication; or archived in an appropriate public repository. The statement needs to describe the availability or the absence of shared data. Authors must include in their statement: a link to the repository they have used, or a statement that it is available as Supporting Information; reference the data in the appropriate sections(s) of their manuscript; and cite the data they have shared in the References section. Whenever possible, the scripts and other artefacts used to generate the analyses presented in the paper should also be publicly archived. If sharing data compromises ethical standards or legal requirements then authors are not expected to share it, but must note this in their statement. For more information, see our Statistics Policy.

- Please include an Abstract Figure file, as well as the Figure Legend text within the main article file. The Abstract Figure is a piece of artwork designed to give readers an immediate understanding of the research and should summarise the main conclusions. If possible, the image should be easily 'readable' from left to right or top to bottom. It should show the physiological relevance of the manuscript so readers can assess the importance and content of its findings. Abstract Figures should not merely recapitulate other figures in the manuscript. Please try to keep the diagram as simple as possible and without superfluous information that may distract from the main conclusion(s). Abstract Figures must be provided by authors no later than the revised manuscript stage and should be uploaded as a separate file during online submission labelled as File Type 'Abstract Figure'. Please also ensure that you include the figure legend in the main article file. All Abstract Figures should be created using BioRender. Authors should use The Journal's premium BioRender account to export high-resolution images. Details on how to use and access the premium account are included as part of this email.

EDITOR COMMENTS

Reviewing Editor:

Comments for Authors to ensure the paper complies with the Statistics Policy (Required):
In parts mean \pm SEM reported. But also confidence intervals are given for this data.

Comments to the Author (Required):

Thank you very much for submitting your manuscript to The Journal of Physiology. Both the reviewer and I found your manuscript substantially improved, and we were interested in the new data on differences in photoreceptor outer segment length. However, Figure 2F should be revised and plotted with the same y-scale as Figure 2E. In addition, please add a legend for the x-axis and provide a brief discussion of why some regions may have longer ROS than others, as this is quite unexpected.

Please also see 'Required Items' above.

REFEREE COMMENTS

Referee #2:

The authors have satisfactorily addressed my major concerns. The new data strengthens the original findings. I have a few minor comments.

Line 222. In line with Nyquist theory it is common practice to sample at twice the frequency of the low pass filter cutoff. Not the other way around. ERG bandwidth is generally in the range of 0.1 to 300-500 Hz.

Line 257. There seems to be a typo or two in the Hill equation. Should it be:

$R = R_{max} / [1 - (I/2/I)^n]$? Please revise.

Line 271. The description of the estimation of cellular noise is unnecessarily convoluted. Please edit.

Figure 2 E and F. Please label axis.

Line 335. Please include the effect-size value and CI to strengthen the argument that there is a structural but non-significant change.

Line 365 Discussion of functional changes in relation to the finding that the single cell recordings do not reflect the reduced responses observed in ERGs of GARP2-KO mice at PM9. An alternative mechanism that the authors may want to consider is that, with degeneration, it is not the currents but the extracellular resistances may be changing in these mice.

END OF COMMENTS

Responses to Referees

Dear Dr Pahlberg,

Re: JP-RP-2025-286548R1-A "Selective Knockout of Murine Glutamic Acid-rich Protein 2 (GARP2) Significantly Alters Dark Continuous Noise in Rod Photoreceptors" by Delores A Stacks, Ulisse Bocchero, Marci L DeRamus, Mai N Nguyen, Jeffrey Messinger, Timothy W. Kraft, Steven Jay Pittler, and Johan Pahlberg

Thank you for submitting your manuscript to The Journal of Physiology. It has been assessed by a Reviewing Editor and by 1 expert referee and we are pleased to tell you that it is acceptable for publication following satisfactory revision.

REQUIRED ITEMS

- You must upload original, uncropped western blot/gel images (including controls) if they are not included in the manuscript. This is to confirm that no inappropriate, unethical or misleading image manipulation has occurred. These should be uploaded as 'Supporting information for review process only'. Please label/highlight the original gels so that we can clearly see which sections/lanes have been used in the manuscript figures. For more information, see: <https://physoc.onlinelibrary.wiley.com/hub/journal-policies#imagmanip>.

For Figure 1 all 3 images are of the entire gel. They are cut to include all markers and all visible bands but exclude the loading lanes. No marks or blemishes were excluded.

- Papers must comply with the Statistics Policy: https://jp.msubmit.net/cgi-bin/main.plex?form_type=display_requirements#statistics.

In summary:

- If $n \leq 30$, all data points must be plotted in the figure in a way that reveals their range and distribution. A bar graph with data points overlaid, a box and whisker plot or a violin plot (preferably with data points included) are acceptable formats.

- If $n > 30$, then the entire raw dataset must be made available either as supporting information, or hosted on a not-for-profit repository, e.g. FigShare, with access details provided in the manuscript.

- 'n' clearly defined (e.g. x cells from y slices in z animals) in the Methods. Authors should be mindful of pseudoreplication.

- All relevant 'n' values must be clearly stated in the main text, figures and tables.

- The most appropriate summary statistic (e.g. mean or median and standard deviation) must be used. Standard Error of the Mean (SEM) alone is not permitted.

- Exact p values must be stated. Authors must not use 'greater than' or 'less than'. Exact p values must be stated to three significant figures even when 'no statistical significance' is claimed.

We have updated all figures and statistics to comply with journal policy. We have added and defined all 'N' values in the main text, figures and tables.

- A Data Availability Statement is required for all papers reporting original data. This must be in the Additional Information section of the manuscript itself. It must have the paragraph heading 'Data Availability Statement'. All data supporting the results in the paper must be either: in the paper itself; uploaded as Supporting Information for Online Publication; or archived in an appropriate public repository. The statement needs to describe the availability or the absence of shared data. Authors must include in their statement: a link to the repository they have used, or a statement that it is available as Supporting Information; reference the data in the appropriate section(s) of their manuscript; and cite the data they have shared in the References section. Whenever possible, the scripts and other artefacts used to generate the analyses presented in the paper should also be publicly archived. If sharing data compromises ethical standards or legal requirements then authors are not expected to share it, but must note this in their statement. For more information, see our Statistics Policy.

We have added a Data Availability Statement to the manuscript.

- Please include an Abstract Figure file, as well as the Figure Legend text within the main article file. The Abstract Figure is a piece of artwork designed to give readers an immediate understanding of the research and should summarise the main conclusions. If possible, the image should be easily 'readable' from left to right or top to bottom. It should show the physiological relevance of the manuscript so readers can assess the importance and content of its findings. Abstract Figures should not merely recapitulate other figures in the manuscript. Please try to keep the diagram as simple as possible and without superfluous information that may distract from the main conclusion(s). Abstract Figures must be provided by authors no later than the revised manuscript stage and should be uploaded as a separate file during online submission labelled as File Type 'Abstract Figure'. Please also ensure that you include the figure legend in the main article file. All Abstract Figures should be created using BioRender. Authors should use The Journal's premium BioRender account to export high-resolution images. Details on how to use and access the premium account are included as part of this email.

EDITOR COMMENTS

Reviewing Editor:

Comments for Authors to ensure the paper complies with the Statistics Policy (Required):
In parts mean \pm -SEM reported. But also confidence intervals are given for this data.

We have updated the text, figures and statistics to comply with journal policy. Please see above.

Comments to the Author (Required):

Thank you very much for submitting your manuscript to The Journal of Physiology. Both the reviewer and I found your manuscript substantially improved, and we were interested in the new data on differences in photoreceptor outer segment length. However, Figure 2F should be revised and plotted with the same y-scale as Figure 2E. In addition, please add a legend for the x-axis and provide a brief discussion of why some regions may have longer ROS than others, as this is quite unexpected.

We have revised Figures 2E and F and added to the Discussion on ROS overgrowth.

Please also see 'Required Items' above.

Please see above.

REFEREE COMMENTS

Referee #2:

The authors have satisfactorily addressed my major concerns. The new data strengthens the original findings. I have a few minor comments.

Line 222. In line with Nyquist theory it is common practice to sample at twice the frequency of the low pass filter cutoff. Not the other way around. ERG bandwidth is generally in the range of 0.1 to 300-500 Hz.

We thank the Reviewer for the observation. This is an editing mistake and has been corrected.

Line 257. There seems to be a typo or two in the Hill equation. Should it be:

$R = R_{max} / [1 - (I/2/I)^n]$? Please revise.

We have corrected the Equation.

Line 271. The description of the estimation of cellular noise is unnecessarily convoluted. Please edit.

The description has been edited.

Figure 2 E and F. Please label axis.

We have updated the axis labels in our new updated Figure 2.

Line 335. Please include the effect-size value and CI to strengthen the argument that there is a structural but non-significant change.

Line 365 Discussion of functional changes in relation to the finding that the single cell recordings do not reflect the reduced responses observed in ERGs of GARP2-KO mice at PM9. An alternative mechanism that the authors may want to consider is that, with degeneration, it is not the currents but the extracellular resistances may be changing in these mice.

We have added some speculation to the Discussion

Dear Dr Pahlberg,

Re: JP-RP-2025-286548R2 "Selective Knockout of Murine Glutamic Acid-rich Protein 2 (GARP2) Significantly Alters Dark Continuous Noise in Rod Photoreceptors" by Delores A Stacks, Ulisse Bocchero, Marci L DeRamus, Mai N Nguyen, Jeffrey Messinger, Timothy W. Kraft, Steven Jay Pittler, and Johan Pahlberg

We are pleased to tell you that your paper has been accepted for publication in The Journal of Physiology.

Yours sincerely,

Katalin Toth
Senior Editor
The Journal of Physiology

IMPORTANT POINTS TO NOTE FOLLOWING ACCEPTANCE OF YOUR PAPER:

- You can help your research get the attention it deserves! Check out Wiley's free Promotion Guide for best-practice recommendations for promoting your work at: www.wileyauthors.com/eoo/guide. You can learn more about Wiley Editing Services which offers professional video, design, and writing services to create shareable video abstracts, infographics, conference posters, lay summaries, and research news stories for your research at: www.wileyauthors.com/eoo/promotion.

- If you would like to receive our 'Research Roundup', a monthly newsletter highlighting the cutting-edge research published in The Physiological Society's family of journals (The Journal of Physiology, Experimental Physiology, Physiological Reports, The Journal of Nutritional Physiology and The Journal of Precision Medicine: Health and Disease), please click this link, fill in your name and email address and select 'Research Roundup': <https://www.physoc.org/journals-and-media/membernews>

EDITOR COMMENTS

Reviewing Editor:

Dear authors,

Thank you for resubmitting your revised manuscript to The Journal of Physiology. The reviewers' concerns are adequately

addressed.

Best wishes

Your Reviewing Editor